# Myeloid-derived suppressor cell mitochondrial fitness governs chemotherapeutic efficacy in hematologic malignancies

Saeed Daneshmandi[1,2], Jee Eun Choi[1], Qi Yan[2], Cameron R. MacDonald[1], Manu Pandey[3], Mounika Goruganthu[2], Nathan Roberts[1], Prashant K. Singh[4], Richard M. Higashi [5], Andrew N. Lane [5], Teresa W-M. Fan[5], Jianmin Wang [6], Philip L. McCarthy [3], Elizabeth A. Repasky[1] & Hemn Mohammadpour [2] ✉

Myeloid derived suppressor cells (MDSCs) are key regulators of immune responses and correlate with poor outcomes in hematologic malignances. Here, we identify that MDSC mitochondrial fitness controls the efficacy of doxorubicin chemotherapy in a preclinical lymphoma model. Mechanistically, we show that triggering STAT3 signaling via β2-adrenergic receptor (β2-AR) activation leads to improved MDSC function through metabolic reprograming, marked by sustained mitochondrial respiration and higher ATP generation which reduces AMPK signaling, altering energy metabolism. Furthermore, induced STAT3 signaling in MDSCs enhances glutamine consumption via the TCA cycle. Metabolized glutamine generates itaconate which downregulates mitochondrial reactive oxygen species via regulation of Nrf2 and the oxidative stress response, enhancing MDSC survival. Using β2-AR blockade, we target the STAT3 pathway and ATP and itaconate metabolism, disrupting ATP generation by the electron transport chain and decreasing itaconate generation causing diminished MDSC mitochondrial fitness. This disruption increases the response to doxorubicin and could be tested clinically.

Hematologic malignancies are cancers of blood-forming tissues. Patients diagnosed with these cancers often receive treatment with combination chemotherapeutic agents. Although the short-term efficacy of chemotherapy is typically driven by its cytotoxic effects on tumor cells, many patients receive additional benefit from the activation of an anti-tumor immune response elicited in response to dying tumor cells[1,2]. This cooperation between chemotherapy agents and anti-tumor immune responses may eliminate or control the proliferation of residual malignant cells not killed by chemotherapy directly, resulting in long-term remission[1]. However, chemotherapy also triggers the proliferation of Myeloid Derived Suppressor Cells (MDSCs), leading to a reduced anti-tumor immune response and diminished effectiveness of the chemotherapy[3].

MDSCs are a heterogeneous group of immature myeloid cells that promote immune escape and tumor growth by inhibiting critical anti-tumor immune responses[4–6]. As a result, high levels of MDSCs are a

[1]Department of Immunology, Roswell Park Comprehensive Cancer Center, Buffalo, New York, NY, USA. [2]Department of Cell Stress Biology, Roswell Park Comprehensive Cancer Center, Buffalo, New York, NY, USA. [3]Department of Medicine, Roswell Park Comprehensive Cancer Center, Buffalo, New York, NY, USA. [4]Department of Cancer Genetics & Genomics, Roswell Park Comprehensive Cancer Center, Buffalo, New York, NY, USA. [5]Department of Toxicology and Cancer Biology, Markey Cancer Center, Center for Environmental and Systems Biochemistry (CESB), Lexington, KY, USA. [6]Department of Biostatistics & Bioinformatics, Roswell Park Comprehensive Cancer Center, Buffalo, New York, NY, USA. ✉e-mail: hemn.mohammadpour@roswellpark.org

negative prognostic marker in patients with hematologic malignancies[7]. Furthermore, preclinical and clinical studies have shown that MDSC depletion, in combination with immunomodulatory therapies including chemotherapeutic agents, can enhance tumor control and treatment efficacy[7].

Doxorubicin, an anthracycline used in cancer treatment, induces oxidative stress by generating reactive oxygen species (ROS) through a reduction process. This oxidative stress causes membrane and DNA damage, triggering apoptotic cell death pathways[8]. MDSCs, often found in oxidative-stress-prone environments like tumors, respond to ROS by activating antioxidative pathways and undergoing transcriptional changes that influence their fate and function[9]. In the tumor microenvironment (TME), MDSCs utilize metabolic reprograming to counter ROS-mediated apoptosis and survive in relatively harsh conditions[10]. This includes the uptake of extracellular nutrients, (e.g., glucose, fatty acids, glutamine, and acetate), in addition to the generation of energy and antioxidant metabolites via glycolysis, the tricarboxylic acid (TCA) cycle, and fatty acid oxidation (FAO)[11]. Recent studies have shown that protumor immune cells, including M2 macrophages and MDSCs, activate central carbon metabolic reprogramming to prevent ROS mediated apoptosis in the TME. Additionally, a shift in central carbon metabolism in MDSCs generates antioxidant metabolites such as phosphoenolpyruvate (PEP), thereby protecting MDSCs from oxidative stress mediated apoptosis[10].

Our previous studies of solid tumors have shown that triggering the β-adrenergic receptor (β-AR), specifically the β2-AR, either through pharmacological agonists such as isoproterenol (ISO) or in vivo via increased adrenergic stress, increasing norepinephrine-dependent signaling, enhances the accumulation and immunosuppressive function of MDSCs[12,13]. However, very little is known regarding how metabolic reprograming induced by β2-AR signaling affects the sensitivity of MDSC to chemotherapy or regulates the efficacy of chemotherapy in hematologic malignancies. In this study, we employed established murine hematologic malignancy models to study the role of β-AR signaling on the metabolic fitness of MDSCs during chemotherapy. Interestingly, we found that the ATP and Itaconate metabolites regulated by STAT-3 signaling are critical regulators of mitochondrial fitness in MDSCs which subsequently determines the efficacy of doxorubicin therapy. Mechanistically, we found that MDSC activation by β-ARs triggers the STAT3 signaling pathway, which ultimately promotes mitochondrial metabolic reprograming and improves MDSC survival due to the acquisition of an anti-apoptotic phenotype. Preclinically, we found that blocking the β2-AR signaling, STAT-3 signaling or ATP and Itaconate metabolites results in enhancement of doxorubicin chemotherapy, marked by reduced tumor growth and increased survival in the EL4 murine lymphoma model.

## Results

### β-Adrenergic receptor (β-AR) signaling modulates doxorubicin efficacy in the EL4 lymphoma model via MDSC regulation

To investigate the role of β-AR signaling in the efficacy of chemotherapy in hematologic malignancies, wildtype (WT) mice were injected intravenously (i.v.) with the lymphoma cell line EL4 followed by a suboptimal dose (day 7; i.v.; 4 mg/kg) of doxorubicin alone or with propranolol (daily; i.p.; 1 mg/kg). The combination of propranolol with doxorubicin increased survival compared to the single agent treatment groups (Fig. 1A). Addition of an anti-Gr1 antibody that depletes Gr-expressing MDSCs enhanced doxorubicin efficacy to that of combined doxorubicin and propranolol treatment. This demonstrated a role for MDSCs in inhibiting chemotherapy-directed tumor control (Fig. 1B). Examination of immune cell compartments from these models demonstrated a significant reduction in MDSC populations in the combination therapy group (Fig. 1C, D). This reduction with combination treatment was accompanied by higher MDSC apoptosis rates, as measured by annexin V staining (Fig. 1E, F), and higher levels of

activated caspase 3 (Fig. 1G, H), and caspase 8 (Fig. 1I, J). This implies that active β-AR signaling promotes MDSC survival and reducing this signaling by β-AR inhibition would decrease MDSC survival. We found that combined doxorubicin and propranolol treatment improved survival in the C1498 acute myeloid leukemia (AML) model (Fig. S1). It is known that MDSCs reduce the function of tumor-reactive adaptive immune cells[7]. We have shown that their depletion by combined modality treatment may encompass another approach to improve the treatment of hematologic malignancies.

Based on our previous data demonstrating that signaling through the β2-AR diminishes the effect of MDSCs during solid tumor therapy, we sought to conditionally knock out the β2-AR in myelomonocytic cells. The gene that encodes the β2-AR in mice is *Adrb2* and mice bearing *Adrb2* deletions are deficient in expressing β2-AR (here we refer to them as β2-AR[-/-])[12,14]. To confirm the key role of β2-AR signaling in MDSCs, we employed β2-AR[-/-] (Fig. 1K, L) and the conditional β2-AR[fl/fl] LysM[Cre] in the myelomonocytic population only (Fig. 1M) mice. Recipients were injected with EL4 (i.v.) and treated with doxorubicin (day 7; i.v.; 4 mg/kg). To examine the role of MDSCs in the β2-AR[-/-] model, anti-Gr1 antibody was injected twice weekly i.p. at 1 mg/kg to deplete MDSCs. Administration of doxorubicin to β2-AR[-/-] mice (Fig. 1K) or β2-AR[fl/fl] LysM[Cre] mice (Fig. 1M) enhanced survival compared to control recipients. Anti-Gr1 administration improved survival of WT recipients receiving doxorubicin similar to that of the β2-AR[-/-] recipient receiving doxorubicin levels (Fig. 1L). These results identify increased MDSC survival, mediated by β2-AR signaling, as an important cellular process limiting chemotherapy efficacy and provide a potential treatment option with β-AR-blockade to be tested in clinical trials to overcome MDSC immunosuppression during treatment.

### β2-AR signaling in MDSCs regulates intrinsic apoptotic pathways

To confirm that β2-AR signaling mediated MDSC resistance to apoptosis, we generated MDSCs in vitro using IL-6 and GM-CSF, and with/without the pan-β-AR agonist, isoproterenol (ISO). MDSCs generated from β2-AR[-/-] mice in the presence or absence of ISO served as additional controls for β-AR specificity. MDSC populations were exposed to increasing concentrations of doxorubicin (1–1000 ng/mL) (Fig. 2). The MDSCs were phenotypically characterized as M-MDSC (CD11b[+]Ly6G[−]Ly6C[hi]) and PMN-MDSC (CD11b[+]Ly6G[+]Ly6C[lo])[15,16]. Detection of apoptosis by annexin V showed that ISO treatment generated resistance to doxorubicin-induced apoptosis in M-MDSCs (Fig. 2A) and PMN-MDSCs (Fig. 2B). This decreased level of apoptosis was associated with lower caspase 3 activity (Fig. 2C, D). Recent advances in single-cell RNA-sequencing (scRNA-seq) offer an opportunity to detect variation at the cellular level and to dissect heterogeneous populations of cells into unique cell clusters. To analyze in vitro generated MDSCs in further detail, we applied scRNA-seq to these cells. We found that in vitro generated MDSCs from the WT or β2-AR[-/-] mice, with or without ISO treatment segregated into six clusters (Fig. 2E, F) which were similar in frequency between treatment groups (Fig. S2). However, when evaluating gene expression profiles, apoptotic pathways showed a lower expression of genes correlating with intrinsic apoptotic pathways in WT + ISO MDSCs compared WT + PBS or β2-AR[-/-] + PBS or +ISO (Fig. 2G). In addition to gene regulation, activation of apoptotic pathways by an apoptotic agent is regulated via post-translational modifications, including protein phosphorylation[17]. Large-scale proteomics studies have identified specific phosphorylation sites on proteins involved in intrinsic or extrinsic apoptotic processes[17,18]. We employed an apoptosis phospho-antibody array system (high-throughput ELISA based antibody array) for qualitative protein phosphorylation profiling of apoptotic pathways (according to the Methods section). Examination of apoptotic

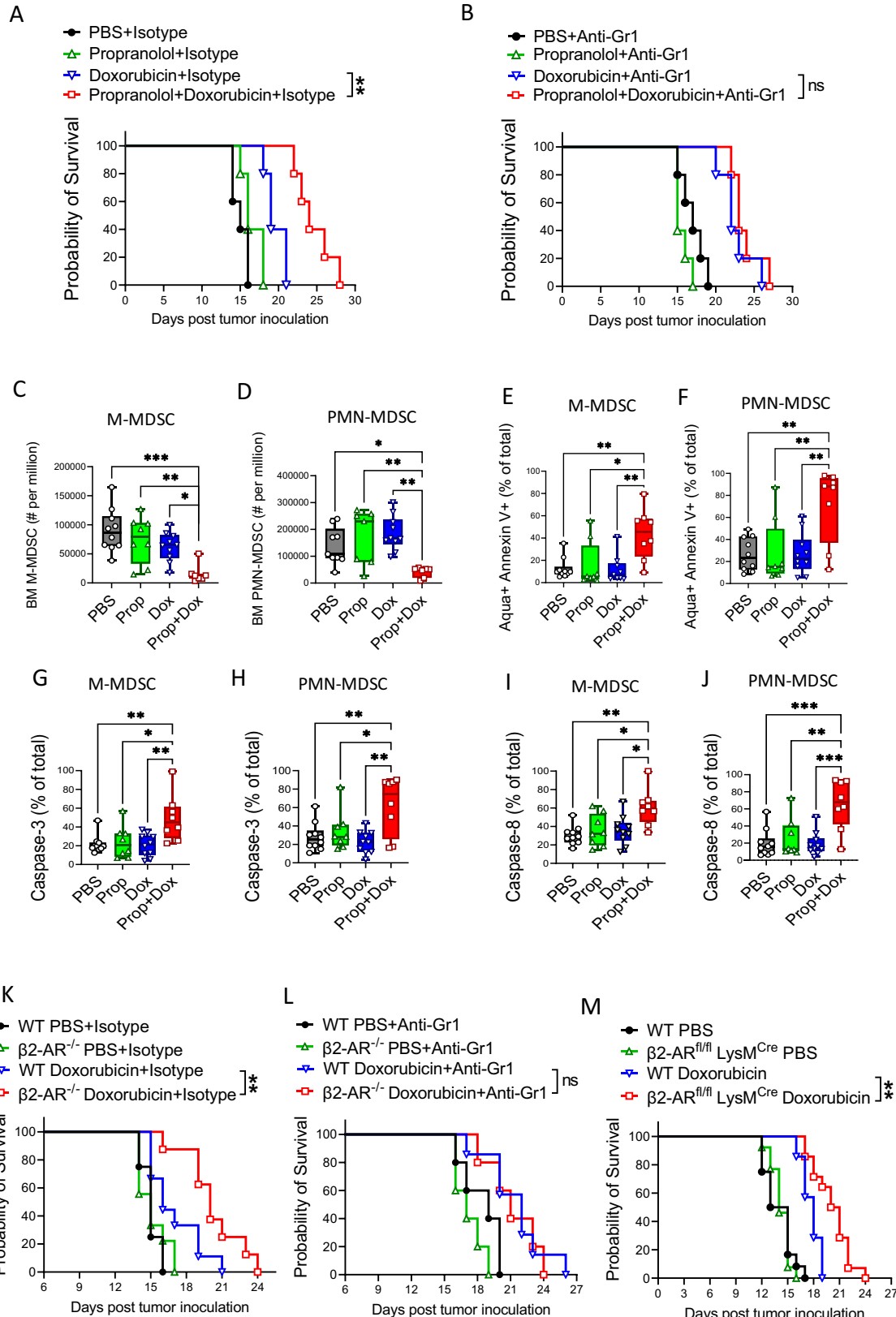

pathways showed upregulation of anti-apoptotic factors such as Ask (p-Ser83)[19], Bad (p-Ser529) and Bcl2 (p-Thr69), and down-regulation of pro-apoptotic factors such as Chk1 (p-Ser286)[20] in WT + ISO compared to WT + PBS control MDSCs after doxorubicin treatment (Fig. 2H). Enrichment of the anti-apoptotic representative marker, Bcl2, post β-AR signaling was confirmed by

western blot analysis (Fig. S3). Comparing the top marker shifts in β2-AR[-/-] + ISO to β2-AR[-/-] + PBS showed no significant change in pro- or anti-apoptotic markers. Potential alterations were seen in signaling markers such intermediates of the NF-κB signaling pathway (Fig. 2I). These data suggest that the effects of ISO on MDSCs is mediated through β2-AR receptor signaling. Activation

**Fig. 1 | Combining adrenergic receptor (β-AR) blockade with doxorubicin results in increased survival of mice with EL4 lymphoma and is associated with changes in MDSC frequency. A, B** The EL4 tumor was given i.v. to mice followed by propranolol (daily; i.p.; 1 mg/kg), or doxorubicin (day 7; i.v.; 4 mg/kg), or both. The mice were treated with (**A**) Isotype (**$p = 0.0026$) or (**B**) anti-Gr1 depleting antibody (twice weekly; i.p.; 1 mg/kg). Two-sided Mantel-Cox log rank test. Frequency of (**C**) CD11b$^+$Ly6C$^+$Ly6G$^-$ (M-MDSC) (*$p = 0.0264$; **$p = 0.0055$; ***$p = 0.0002$) and (**D**) CD11b$^+$Ly6C$^+$Ly6G$^+$ (PMN-MDSC) (*$p = 0.0467$; **$p = 0.0012$; **$p = 0.0075$) in bone marrow of mice treated as in (**A**). Apoptosis rate in (**E**) M-MDSCs (*$p = 0.0110$; **$p = 0.0025$; **$p = 0.0025$) and (**F**) PMN-MDSCs (**$p = 0.0026$; **$p = 0.0028$; **$p = 0.0062$) from (**A**) measured by Annexin V staining (apoptotic cell: Aqua$^+$ Annexin V$^+$). Activated levels of caspase 3 determined by CaspGLOW™ Caspase-3 staining on (**G**) M-MDSCs (*$p = 0.0195$; **$p = 0.0065$) and (**H**) PMN-MDSCs

(*$p = 0.0493$; **$p = 0.0014$; **$p = 0.0068$) from mice treated as in (**A**). Activated levels of caspase 8 determined by CaspGLOW™ Caspase-8 staining on (**I**) M-MDSCs (*$p = 0.0114$; *$p = 0.0169$; **$p = 0.0016$) and (**J**) PMN-MDSCs (**$p = 0.0026$; ***$p = 0.002$; ***$p = 0.003$) from mice treated as in (**A**). In (**C**–**J**) data presented as mean ± SEM. Pool of two independent experiments with $n = 5$ mice per group. Survival of EL4 tumor bearing mice in WT vs. β2AR$^{-/-}$ mice treated with doxorubicin (day 7; i.v.; 4 mg/kg) and treated with (**K**) Isotype (**$p = 0.0019$) or (**L**) anti-Gr1 depleting antibody (twice weekly; i.p.; 1 mg/kg). Two-sided Mantel-Cox log rank test. **M** Survival of EL4 tumor bearing recipients in WT, β2-AR$^{-/-}$ and the conditional β2-AR$^{fl/fl}$ LysM$^{Cre}$ myelomonocytic population knockouts. WT compared to β2-AR$^{fl/fl}$-LysM$^{Cre}$ mice treated with doxorubicin (day 7; i.v.; 4 mg/kg) (**$p = 0.0064$). Two-sided Mantel-Cox log rank test. Source data containing independent experiments raw data are provided as a Source Data file.

---

of both Bcl2 and Bcl-xL protect cells from apoptosis via mROS[21]. Apoptosis signal-regulating kinase 1 (Ask1), plays essential roles in stress-induced apoptosis and its kinase activity is inhibited by phosphorylation at Ser83 by Akt[19]. Bad is a pro-apoptotic member of the Bcl-2 family and survival factors inhibit the apoptotic activity of Bad by phosphorylation at Ser112[22]. Elevated ROS levels also activate the ATR-Chk1 axis[23] and Chk1 phosphorylation is correlated with higher DNA damage[20]. These results confirm the key role β-AR signaling to suppress intrinsic apoptotic pathways, inducing resistance to doxorubicin-mediated MDSC apoptosis.

## Activation of MDSC STAT3 signaling regulates mitochondrial metabolism to generate ATP

The intrinsic apoptosis pathway, regulated by mitochondrial metabolism, was downregulated in MDSCs after β-AR signaling (Fig. 2). We hypothesized that activation of β-AR signaling in MDSCs controls their mitochondrial metabolism. Seahorse metabolic flux assays showed that WT + ISO MDSCs experienced a significantly higher rate of mitochondrial respiration compared to controls, as measured by an increased basal oxygen consumption rate (OCR) rate (Fig. 3A). The effect appears to be mediated by the β2-AR as ISO treatment in the β2-AR$^{-/-}$ cells is the same as WT cells treated with PBS. The increase in MDSC mitochondrial respiration accompanied an ~35% increase in cellular ATP generation (Fig. 3B). Higher mitochondrial metabolism occurred with enhanced expression of TCA cycle-associated genes (Fig. 3C). Thus, β2-AR signaling may serve as a rheostat of MDSC mitochondrial respiration. Additional examination of MDSC subpopulations demonstrated that CD11b$^+$Ly6C$^+$Ly6G$^-$ (M-MDSC) had higher mitochondrial respiration capacity than the CD11b$^+$Ly6C$^-$Ly6G$^+$ (PMN-MDSC) subpopulation, although both subpopulations showed enhanced mitochondrial respiration post β-AR signaling (Fig. S4A–C).

M-MDSCs and PMN-MDSCs share key biochemical features that enable their suppression of immune responses, including the upregulation of signal transducer and activator of transcription 3 (STAT3) expression[4]. During differentiation, cytokines including IL-6 and GM-CSF induce activation of STAT3 signaling[5,24]. Activated STAT3 signaling during MDSC activation correlated with elevated expression of gene patterns related to Janus kinase (JAK) and STAT signaling[25]. Screening using a cell signaling phospho-antibody array demonstrated enhanced JAK/STAT pathway activity in MDSCs after β2-AR signaling (Fig. 3D) compared to controls. To confirm this, we generated MDSCs in vitro with and without ISO stimulation and assessed STAT3 activation by determining STAT3 phosphorylation (Tyr705). JSI-124 (cucurbitacin I) is a cell permeable molecule that inhibits tyrosine kinase activity that is specific to JAK/STAT3 signaling[26]. In parallel studies, cells were treated with JSI-124 (1 μM) on day 3 of MDSC generation. Western blot analysis confirmed enhancement of pSTAT3, which was blocked by JSI-124 treatment recapitulating the effect seen with the β2-AR$^{-/-}$ cells (Fig. 3E, F). This corroborates our previous report demonstrating

enhanced MDSC STAT3 signaling via activation of β-AR signaling in vivo[12].

To clarify the role of the STAT3 signaling pathway in the shift in mitochondrial respiration post β2-AR activation, we triggered β2-AR using ISO, exposing MDSCs to increasing JSI-124 (0.5–3 μM) concentrations. We utilized the Agilent Seahorse XF ATP Real-Time rate assay to determine the ATP production rates from glycolysis and oxidative phosphorylation simultaneously. The mitochondrial respiration rate was decreased by JSI-124 in a dose-dependent manner, along with decreased ATP synthesis and reserve capacity (Fig. 3G). Although glycolysis generated ATP more rapidly than oxidative phosphorylation, glycolysis showed a much lower response to JSI-124 (Fig. 3H). These studies show that STAT3 signaling regulates mitochondrial ATP generation but not glycolytically-derived ATP generation.

We investigated mechanisms of how STAT3 signaling in MDSCs regulates mitochondrial ATP generation. During mitochondrial ATP generation, the electron transport chain (ETC) contains two electron transport pathways: Complex I/III/IV with NADH as the substrate, and complex II/III/IV, using succinate as the substrate. Mitochondrial complex V or F1-ATP synthase catalyzes ATP synthesis driven by the mitochondrial proton gradient generated by the ETC[27,28]. The physiological regulator of the ATP synthase is ATPase inhibitory factor 1 (ATPIF1), a 12 kD protein, that inhibits both the synthetic and hydrolytic activities of the enzyme[29,30]. We hypothesized that ATPIF1 could be a potential regulator of ATP generation in ISO-treated MDSCs. We determined ATPIF1 expression in WT + ISO and controls by western blot (Fig. 3I, J), which demonstrated lower levels of ATPIF1 in WT + ISO compared to control MDSCs. After activating β-AR signaling in MDSCs, blocking STAT3 signaling by JSI-124 results in upregulation of ATPIF1 expression (Fig. 3K). To confirm the suppressive function of pSTAT3 on *Atpif1* expression, we assessed the direct interaction of pSTAT3 and STAT3 with the promoter region of *Atpif1* by the Chromatin Immunoprecipitation quantitative PCR (ChIP-qPCR) assay. Treatment of MDSCs with ISO decreased the binding of STAT3 and pSTAT3 to *Atpif1* promoter regions in two of the 3 potential binding sites (from matching consensus sequences generated by the UCSC Genome Browser on Mouse [GRCm39/mm39] program) (Figs. 3L and S5). In non-treated MDSCs, STAT3 binds to *Atpif1* promotor at higher rate compared to pSTAT3 (Fig. 3L). This confirmed the decrease in STAT3 binding on *Atpif1* promoter post β-AR signaling to reduce *Atpif1* gene expression. STAT3 signaling is key for shaping MDSC differentiation and function. This includes regulating the gene expression profile toward a suppressive phenotype and functions. Although pSTAT3 regulates gene expression at early time points post activation, unphosphorylated STAT3 is responsible for pSTAT3 non-responding genes at later time points[31]. Unphosphorylated STAT3 accumulates in response to gp130-linked cytokines (e.g., IL-6), activating transcription. Unphosphorylated STAT3 sustains cytokine-dependent signaling at later times through a distinct mechanism from that used by pSTAT3.

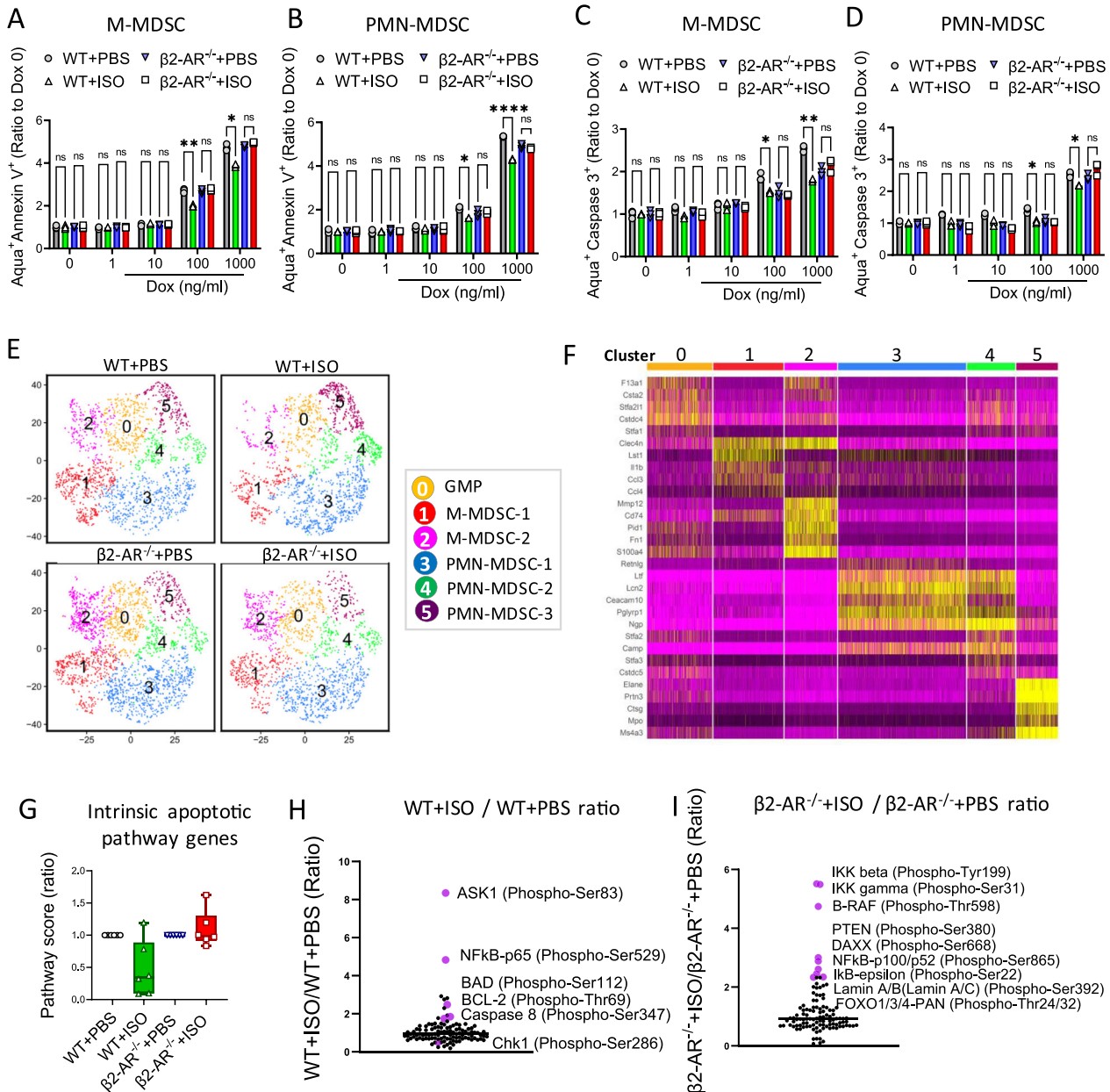

**Fig. 2 | Triggering β-AR signaling suppresses intrinsic apoptotic pathways in MDSCs. A**, **B** MDSCs were generated in vitro using IL-6 (40 ng/ml) and GM-CSF (40 ng/ml) with and without isoproterenol (10 μM) treatment in both wild type (WT) and β2-AR⁻/⁻ MDSCs. These cells were then exposed to increasing concentrations of doxorubicin (1–1000 ng/ml). After 20 h. the apoptosis rate was measured by Annexin V (apoptotic cell: Aqua+ Annexin V + ) in (**A**) CD11b⁺Ly6C⁺Ly6G⁻ (M-MDSC) (*$p$ = 0.0109; **$p$ = 0.0030) and (**B**) CD11b⁺Ly6C⁺Ly6G⁺ (PMN-MDSC) (*$p$ = 0.0216; ****$p$ < 0.0001). One-way ANOVA with Bonferroni's multiple comparison tests. Data are presented as mean ± SEM. Representative of three independent experiments with $n$ = 3 technical replicates per group. Activated caspase 3 was detected in in vitro generated MDSCs as (**C**) M-MDSC (*$p$ = 0.0419; **$p$ = 0.0019) and (**D**) PMN-MDSC (*$p$ = 0.0158; *$p$ = 0.0455) by CaspGLOW™ Caspase-3 staining assay. One-way ANOVA with Bonferroni's multiple comparison tests. Data are

presented as mean ± SEM. Representative of three independent experiments with $n$ = 3 technical replicates per group. **E**, **F** MDSCs were generated in vitro, and their transcriptomics profile was evaluated by single cell RNA sequencing (scRNA-seq). **E** tSNE plots demonstrate distribution of MDSC subpopulations, and (**F**) the heatmap summarizes the top genes correlated with each subpopulation. **G** Overall gene profile shift in generated MDSCs for intrinsic apoptotic pathway. Each data point represents pathway score value of a cluster (clusters 0–5 of **E**, **F**). Results are shown as representative of $n$ = 6 clusters per group. Data are presented as mean ± SEM. Apoptosis Phospho-Antibody Array used to detect apoptotic pathways on in vitro generated (**H**) WT + ISO vs. WT + PBS, and (**I**) β2-AR⁻/⁻ + ISO vs. β2-AR⁻/⁻ + PBS control MDSCs. Source data containing independents experiment raw data are provided as a Source Data file.

To determine how the STAT3 pathway post β2-AR signaling reduces *Atpif1* expression (a physiological need for MDSC-enhanced OXPHOS), we performed ChIP-qPCR to confirm that STAT3/pSTAT3 binds to the *Atpif1* promotor post AR signaling. Our results show that in untreated MDSCs, unphosphorylated STAT3 is attached to the *Atpif1* promoter. Post β2-AR stimulation, STAT3 phosphorylation (pSTAT3) is enhanced

resulting in reduced STAT3 availability and decreased binding to the *Atpif1* promoter, suppressing *Atpif1* expression. This is in accordance with the subtle and sustained stimulation of the STAT3 pathway at physiological levels in MDSCs relying on higher OXPHOS. These results link activated STAT3 signaling post β-AR activation to enhanced mitochondrial respiration and increased ATP generation mediated by

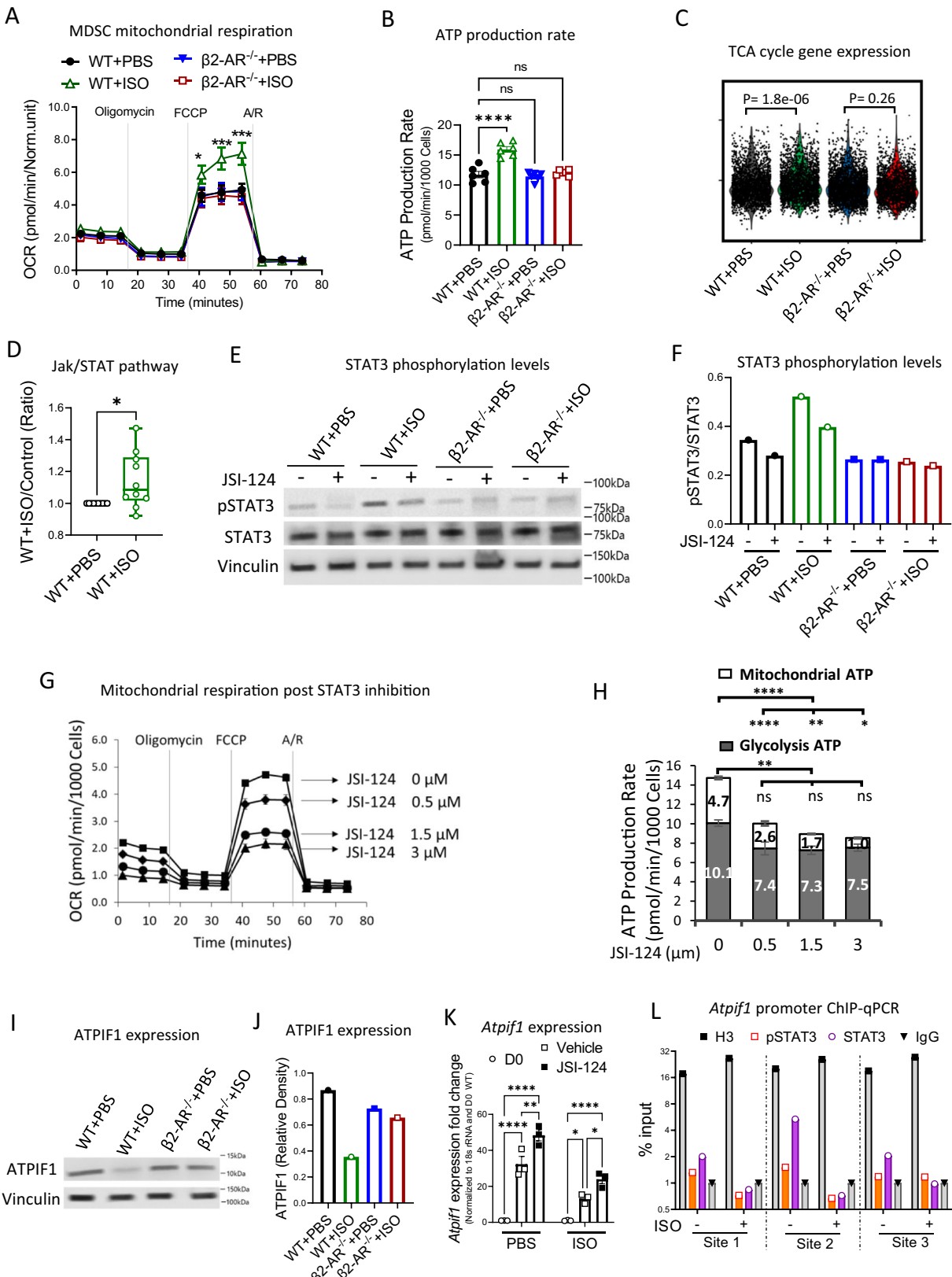

the suppression of ATPIF1. Although previous studies have reported that whole-body deletion of ATPIF1 caused no obvious phenotype in non-stressed mice[32], ATPIF1 is a known regulator of mitochondrial respiration in cancer, hepatic, cardiac and neurological cells/tissues after hypoxic or adrenergic stimulation[33–35]. β-AR signaling also controls ATPIF1 function to upregulate ATP production in the heart,

thereby regulating cardiac dysfunction[36,37]. These results together point to a key role for ATPIF1 activation post STAT3 signaling in MDSC metabolism. However, the shift of ATPIF1 function would not be the only regulator of mitochondrial respiration and other metabolic reprograming processes also could contribute to regulation of MDSC mitochondrial respiration.

**Fig. 3 | β-AR signaling in MDSCs induces higher mitochondrial ATP generation, correlating with regulation of ATPIF1. A, B** MDSCs were generated in vitro as in Fig. 2. **A** Mitochondrial respiration (*$p = 0.0165$; ***$p < 0.001$), **B** ATP generation (****$p < 0.0001$) determined by Seahorse assay. One-way ANOVA with Bonferroni's multiple comparison tests. Data presented as mean ± SEM. Representative of two independent experiments with $n = 4$ technical replicates per group.
**C** Transcriptomics profile of TCA cycle gene pool determined in scRNA-seq as Fig. 2E, F. **D** Activation of Jak-STAT pathway by ISO treatment of MDSCs detected by cell signaling phospho-antibody array. Phosphorylation levels for pSTAT3 (Tyr705), pSTAT2 (Tyr690), pSTAT5B (Ser731), pSTAT6 (Thr645), pJAK2 (Tyr1007), pJAK2 (Tyr221), pAKT1 (Ser473), pAKT1S1 (Thr246), pAKT2 (Ser474), pRaf1 (Ser259) (*$p = 0.0292$). The two-sided paired t-tests. Data are presented as mean ± SEM. $n = 10$ target protein per group. **E, F** MDSCs were generated as in (**A**), and treated with pSTAT3 blocker, JSI-124 (1 μM) on days 3. **E** Western blot image shows the total STAT3 and pSTAT3 (Tyr705) and (**F**) quantified as pSTAT3/STAT3 ratio. $n = 1$ per

group. **G** Mitochondrial respiration in MDSCs treated with increasing concentrations of the STAT3 blocker, JSI-124 (1 μM). $n = 3$ replicates per group.
**H** Mitochondrial and glycolysis-based ATP generation was assessed using the ATP assay kit on MDSCs treated with increasing concentration of the STAT3 blocker, JSI-124. $n = 3$ replicates per group. **I, J** ATPIF1 expression in generated MDSCs was determined by western blot. $n = 1$ per group. **K** *Atpif1* expression in WT + ISO and WT + PBS control MDSCs post STAT3 signaling blockade by JSI-124 (1 μM) (*$p = 0.0109$; *$p = 0.0252$; **$p = 0.0016$; ****$p < 0.0001$). Two-way ANOVA with Tukey's multiple comparison tests. Data are presented as mean ± SEM. Representative of two independent experiments with $n = 3$ technical replicates per group.
**L** ChIP-qPCR assay demonstrated STAT3 compared to pSTAT3 binding to the *Atpif1* promoter at two of the 3 potential binding sites after stimulation with isoproterenol (10 μM) for 30 min. Anti-histone H3 (H3): positive control; Anti-IgG isotype: negative control. Source data containing independent experiments raw data are provided as a Source Data file.

## STAT3 signaling in MDSCs downregulates AMP-activated protein kinase (AMPK) signaling by increasing the ATP/AMP ratio

Viable cells maintain a high, nonequilibrium ratio of ATP to ADP for survival. Due to the adenylate kinase reaction (2ADP ↔ ATP + AMP), AMP rises when the ATP:ADP ratio falls. Increased AMP activates the AMPK as a major sensor of cellular energy homeostasis[38]. AMPK also regulates cell apoptosis through several mechanisms involving the induction of metabolic reprograming[39,40]. Activated AMPK promotes catabolic pathways that produce ATP while inhibiting anabolic pathways that consume ATP, including macromolecule biosynthesis and cell proliferation[41]. To uncover the molecular mechanisms responsible for apoptotic pathway control resulting from shifted ATP generation, we examined the alteration in the AMPK signaling pathway in β-AR (STAT3) stimulated MDSCs.

As seen in (Fig. 3G, H), Seahorse analysis showed higher mitochondrial ATP generation in MDSCs post β-AR mediated STAT3 signaling. To detect ATP and AMP levels in MDSCs, we performed stable isotope-resolved metabolomics (SIRM) analysis to detect the transformation of $^{13}C_5$-glutamine into ATP and AMP over 20 h. Figure 4A, B show the responses of unlabeled metabolites (0), $^{13}C$ incorporation from glutamine (C* = sum of $^{13}C_1$ to $^{13}C_x$), and total metabolite content (Total = sum of unlabeled and isotope labeled). Each isotopologue was quantified and expressed as μmol/g protein. The SIRM results showed little change in ATP (Fig. 4A) but significantly lower AMP (Fig. 4B) production in WT + ISO compared to WT + PBS MDSCs. The cellular ATP/AMP ratio increased during the 20 h culture in WT + ISO compared to WT + PBS (Fig. 4C). There was no significant change in ATP and AMP concentrations in β2-AR$^{-/-}$ + ISO compared β2-AR$^{-/-}$ + PBS MDSCs (Fig S6A–C). As an energy sensor, the response of AMPK to ATP/AMP shifts is regulated by two mechanisms; (1) AMPK expression is controlled by upstream signaling pathways or (2) canonical AMP-dependent mechanisms control the AMPK enzyme via phosphorylation at various locations including the active site at Thr172[38,42]. To correlate the effects of β2-AR-induced STAT3 signaling with regulation of the AMPK pathway, we assessed the effect of STAT3 blockade on AMPK expression and AMPK phosphorylation. Our phospho-antibody array data showed higher activation of AMPK pathway markers in WT + ISO MDSCs (Fig. 4D). Western blot analysis demonstrated increased expression of pAMPK (Tyr172) after blocking of STAT3 signaling in WT + ISO and control MDSCs (Fig. 4E, F). The level of total AMPK protein showed a small increase by STAT3 blockade in WT + ISO MDSC (Fig. 4E). Regulatory mechanisms on AMPK expression have been seen in cancer cells along with key signaling molecules such as mammalian Target Of Rapamycin (mTOR) (Raptor) or Hypoxia-Inducible Factor 1-alpha (HIF-1α) which control AMPK protein expression. This leads to the induction of subsequent metabolic reprogramming[43]. These results confirm the suppression of the AMPK pathway after β2-AR-induced STAT3 signaling and the induction of a higher MDSC ATP/AMP ratio.

## Enhanced mitochondrial TCA cycle activity with the transformation of glutamine into itaconate

For ATP generation in mitochondria, key metabolic sources include glucose and glutamine that enter cells via Glut1 and amino acid transporters (SNAT1 [Slc1a5] and SNAT2 [Slc38a2]), respectively. Glucose is mainly oxidized via cytoplasmic glycolysis to pyruvate which enters the TCA cycle via pyruvate dehydrogenase (PDH) (AcCoA) or pyruvate carboxylase (PC) (forming oxaloacetate) to form citrate. Glutamine is also a major source fueling the TCA cycle, which is successively deamidated to glutamate and transaminated to alpha-ketoglutarate which can enter the TCA cycle. This transformation is an anaplerotic input into the TCA cycle to fuel mitochondrial energy production and anabolic metabolism. Glutamine is also a glucogenic amino acid whose carbon can participate in gluconeogenesis via phosphoenolpyruvate carboxykinase but also generates pyruvate via malic enzyme activity[44,45].

Seahorse metabolic flux assay showed a higher rate of mitochondrial respiration and mitochondrial ATP generation post STAT3 signaling (Fig. 3G, H). We then employed SIRM analysis to track the transformation of $^{13}C_5$-glutamine into TCA cycle (Fig. 5A) or glycolytic (Fig. S7) intermediates. MDSCs were generated in vitro and treated simultaneously with $^{13}C_5$-glutamine, then examined for metabolic incorporation of the heavy isotopes at 2 h and 8 h. Figure 5A shows the amount of unlabeled (0), $^{13}C$ incorporation from glutamine (C* = sum of $^{13}C1$ to $^{13}C_x$), and total metabolite components (sum of unlabeled and isotope labeled) in μmole/g protein. Glycolytic pathway intermediates were not significantly different in WT + ISO and WT + PBS treated MDSCs (Fig. S7A, B). However, there were significant differences in TCA cycle metabolites (Fig. 5A). There was no significant increase in TCA cycle intermediates (Fig. S6D) or changes in glycolytic (Fig. S7C) metabolites of β2-AR$^{-/-}$ + ISO compared to β2-AR$^{-/-}$ + PBS groups. These results parallel the Seahorse assay results, demonstrating no difference between WT + ISO MDSCs and control MDSCs in glycolytic rate (Fig. S7D) and glycolysis-based ATP generation (Fig. S7E).

During MDSC differentiation from bone marrow, GM-CSF and IL-6 signaling induce STAT3 signaling[5,24] upregulating glutamine uptake[46]. We observed that glutamine was the preferred fuel source for the TCA cycle via glutaminolysis catalyzed by glutaminase (GLS) (Fig. 5A). β-AR signaling led to increased glutamine uptake and transformation into glutamate (Fig. 5A-a), followed by enhanced incorporation of glutamine-derived carbon in all examined TCA cycle metabolites including alpha-ketoglutarate (αKG) (Fig. 5A-b), succinate (Fig. 5A-c), fumarate (Fig. 5A-d) and malate (Fig. 5A-e), citrate (Fig. 5A-f), and isocitrate (Fig. 5A-g). Furthermore, itaconate and itaconate producing metabolites (cis-aconitate) increased to higher levels than TCA cycle intermediates and β-AR signaling boosted the TCA cycle to generate enhanced itaconate production via cis-aconitate (Fig. 5A h-i). Comparative metabolite examination showed that glutamine transforms

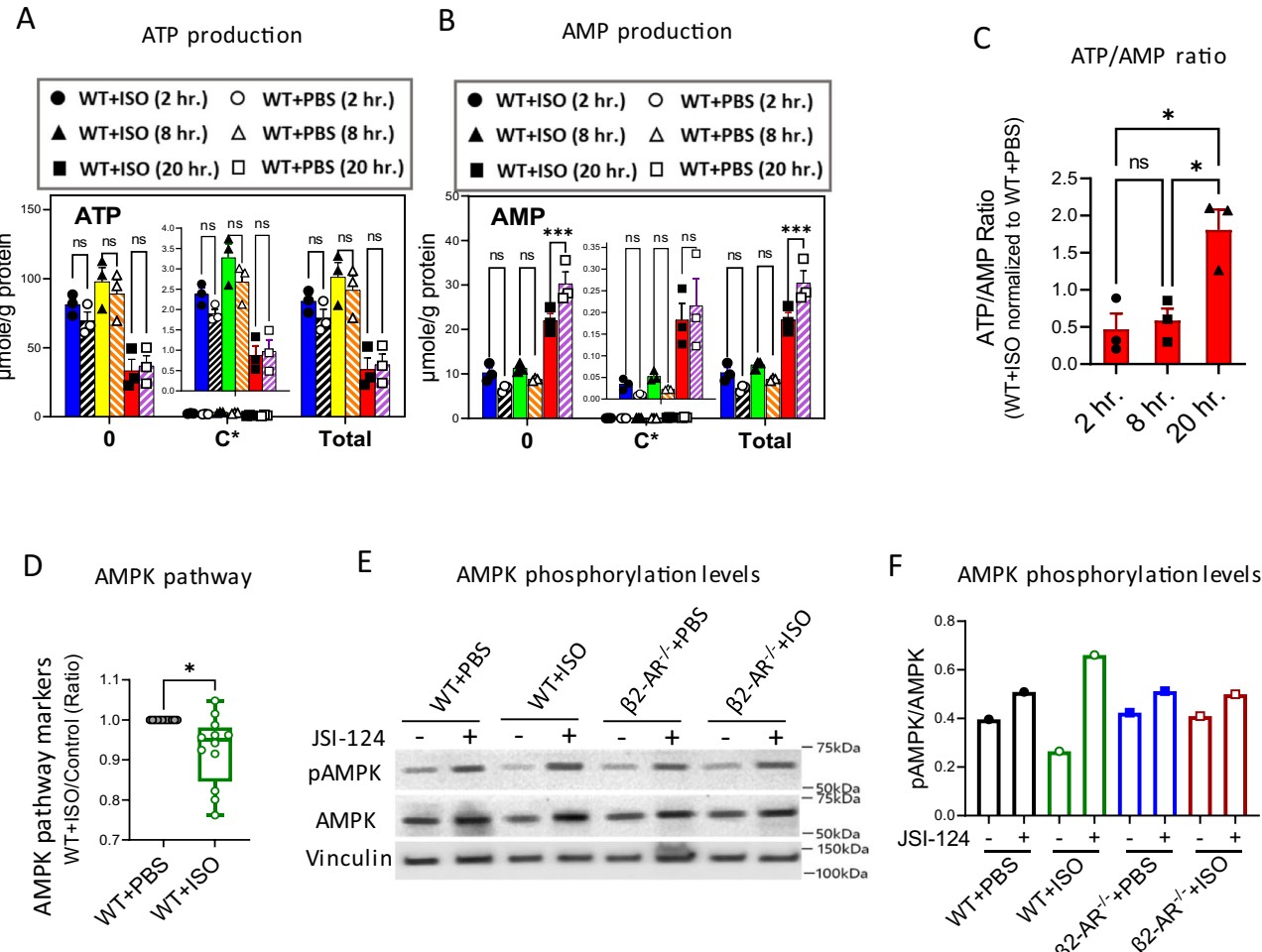

**Fig. 4 | STAT3 signaling induced by stimulation of β-AR increases the ATP/AMP ratio, resulting in downregulation of AMPK signaling in MDSCs.** In vitro generated WT + PBS and WT + ISO MDSCs as in Fig. 2 were treated with treated with $^{13}C_5$-Gln and isotope-labeling levels in (**A**) ATP and (**B**) AMP were compared at 2 h, 8 h and 20 h. post treatment by Ion Chromatography-Ultra High-Resolution Mass Spectrometry (IC-UHRMS) as unlabeled (0), isotope labeled (C*), and total (pool of 0 + C*) (***$p = 0.0001$; ***$p = 0.0002$). Two-way ANOVA with Tukey's multiple comparison tests. Data are presented as mean ± SEM. Representative of two independent experiments with $n = 3$ technical replicates per group. **C** The ATP/AMP ratio in WT + ISO compared WT + PBS control MDSCs was calculated based on the total ATP and AMP levels in the tracing experiment. (*$p = 0.0120$; *$p = 0.0182$) Data are presented as mean ± SEM. Representative of two independent experiments with $n = 3$ technical replicates per group. **D** Activation levels of signaling markers related

to the AMPK pathway were determined post ISO treatment treated (WT + ISO) and control (WT + PBS) MDSCs using phospho-antibody array. Phosphorylation levels were measured for pAMPKbeta1 (Ser182), pMAP3K7/TAK1 (Thr184), p4E-BP1 (Thr45), pACC1 (Ser80), pP70S6K (Thr421), pP70S6K (Ser424), pPDK1 (Ser241), pCyclin D1 (Thr286), pEEF2 (Thr56), pelF4E (Ser209), pPP2A-alpha (Tyr307), pFOXO1/3/4-pan (Thr24/32). (*$p = 0.0124$) The paired $t$ tests. Data are presented as mean ± SEM. $n = 12$ target protein per group. **E, F** Western blot analysis demonstrating AMPK and pAMPK (Tyr172) levels in generated MDSCs, treated with the STAT3 signaling blocker, JSI-124 (1 μM). Derived from the same samples in Fig. 3E, F. Results are representative of three independent experiments with $n = 1$ per group. Source data containing independent experiments raw data are provided as a Source Data file.

through the TCA cycle and accumulates mostly in the form of itaconate, cis-aconitate and citrate in WT + ISO MDSCs compared to control (Fig. 5A, B).

Based on these results, we hypothesized that itaconate could exert a major effect on β-AR signaling in MDSCs. Itaconate has been recently reported as an important regulator of myeloid cell immune responses (e.g., M2 macrophages)[47]. However, the effect of itaconate on MDSC function and survival has not been investigated in detail. Itaconate is generated from cis-aconitate by the mitochondria-associated enzyme aconitate decarboxylase 1 (Acod1; also known as CAD), originally called immune responsive gene 1 (Irg1). Macrophage expression of this enzyme increases over toll-like receptor (TLR) ligands including LPS and cytokines, such as type I and type II interferons[47]. Acod1 is upregulated in murine and human tumor-infiltrating neutrophils and generated itaconate enhances neutrophil

resistance to ferroptosis and stimulates breast cancer metastasis via NF-E2–related factor 2 (Nrf2)[48]. We found higher *Acod1* expression in WT + ISO MDSCs compared to controls (Fig. 5C). Itaconate generation regulates ROS levels in macrophages[47] and tumor neutrophils[48]. Since we observed higher mitochondrial respiration but lower mROS generation (Fig. 5D, E) in MDSCs post β2-AR signaling, we hypothesized that itaconate could be a major metabolite regulating mROS in MDSCs. We then generated MDSCs from WT and Acod1[-/-] mice in presence of ISO and β2-AR[-/-] MDSCs as controls and exposed them to increasing doxorubicin concentrations. Apoptosis by Annexin V assay indicated that MDSCs generated in presence of ISO, but lacking the Acod1 enzyme had a higher apoptosis rate than WT + ISO; comparable to the WT + PBS control (Fig. 5F, G). This confirms the role of Acod1 and itaconate in apoptosis regulation in both M-MDSCs (Fig. 5F) and PMN-MDSCs (Fig. 5G). Moreover, Acod1[-/-] MDSCs were able to limit

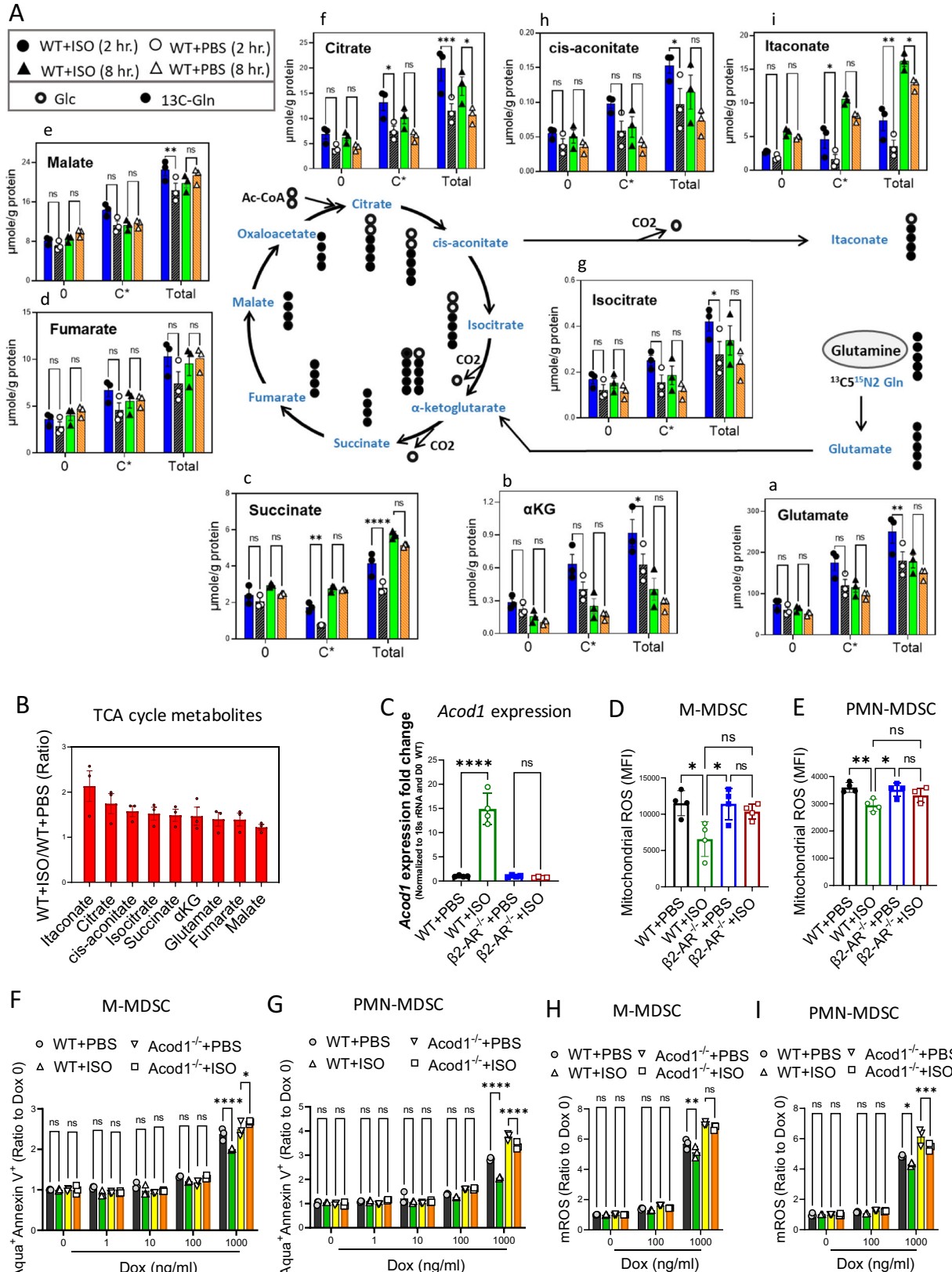

doxorubicin-induced mROS in both M-MDSCs (Fig. 5H) and PMN-MDSCs (Fig. 5I). This confirms the recent study demonstrating decreased apoptosis and increased neutrophil supported tumor viability due to the action of Acod1 and itaconate generation[48]. Our results confirm that β2-AR signaling induces MDSC mitochondrial metabolic reprograming, enhancing TCA cycle activity and itaconate generation which inhibit doxorubicin-induced apoptosis.

**Itaconate controls doxorubicin-induced oxidative stress by regulation of Nrf2 expression and antioxidant pathway activation**

Itaconate, a TCA cycle product is generated in the mitochondrial matrix, and has modest cell permeability. 4-octyl itaconate (4OI) is a cell-permeable itaconate derivative which is converted to itaconate

**Fig. 5 | Enhanced transformation of glutamine to itaconate via the TCA cycle by β-AR signaling suppresses mROS generation, enhancing resistance to doxorubicin in MDSCs. A, B** MDSCs were generated as in Fig. 4 and treated with $^{13}C_5$-Gln and isotope-labeling levels into TCA cycle metabolites were examined at 2 h and 8 h. timepoints. **A** Glutamine transformation into TCA cycle metabolites and accumulation in itaconate are shown. **B** Accumulation levels of TCA cycle metabolites at the 8 h timepoint in WT + ISO compared WT + PBS control MDSCs. (**a:** **$p = 0.0097$; **b:** *$p = 0.0325$; **c:** **$p = 0.0021$; ****$p < 0.0001$; e: **$p = 0.0077$; f: *$p = 0.0228$; ***$p = 0.0002$; g: *$p = 0.0471$; h: *$p = 0.0295$; i: *$p = 0.0441$; *$p = 0.0480$; **$p = 0.0078$). Two-way ANOVA with Tukey's multiple comparison tests. Data presented as mean ± SEM. Representative of two independent experiments with $n = 3$ technical replicates per group. 0 = unlabeled; C* = sum of $^{13}C$-labeled species with 0-x number of $^{13}C$; Total = sum of all the unlabeled and isotope labeled species. **C** Expression of *Acod1* was measured by real-time PCR in generated MDSCs (****$p < 0.0001$). Mitochondrial ROS (mROS) was detected by MitoSox Red on (**D**) CD11b+Ly6C+Ly6G- (M-MDSC) and (*$p = 0.0141$; *$p = 0.0164$) (**E**) CD11b+Ly6C+Ly6G+ (PMN-MDSC) (*$p = 0.0201$; **$p = 0.0080$). In (**C–E**) One-way ANOVA with Bonferroni's multiple comparison tests. Data presented as mean ± SEM. Representative of two independent experiments with $n =$ at least 3 technical replicates per group. **F, G** MDSCs from WT and ACOD1−/− mice were generated in vitro with and without isoproterenol (10 μM) treatment and exposed to increasing concentrations of doxorubicin (0–1000 ng/ml). Apoptosis (Aqua+ Annexin V+) was measured by Annexin V staining in (**F**) M-MDSCs (*$p = 0.0498$; ****$p < 0.0001$) and (**G**) PMN-MDSCs (****$p < 0.0001$). mROS levels were examined in (H) M-MDSCs (*$p = 0.0019$) and (**I**) PMN-MDSCs (*$p = 0.0140$; ***$p = 0.009$). In (**F–I**) Two-way ANOVA with Tukey's multiple comparison tests. Data presented as mean ± SEM. Representative of two independent experiments with $n = 3$ technical replicates per group. Source data containing independent experiments raw data are provided as a Source Data file.

intracellularly by esterase activity[49]. To confirm the effect of itaconate effect on MDSC metabolic fitness described above, MDSCs were generated in vitro for 4 days, pre-treated with 4OI (0, 125, and 250 μM) for 4 h, followed by exposure to increasing doxorubicin concentrations. Annexin V staining showed a lower apoptosis rate with 4OI treatment of M-MDSCs (Fig. 6A) and PMN-MDSCs (Fig. 6B). CaspGLOW™ Caspase-3 staining assay demonstrated lower active caspase 3 levels, after 4OI treatment of M-MDSCs (Fig. 6C) and PMN-MDSCs (Fig. 6D). We applied phospho-proteomics to analyze apoptotic pathways using an apoptosis phospho-antibody array system on in vitro generated MDSCs, pre-treated with 250 μM of 4OI for 4 h, then exposed to 100 nM of doxorubicin for 20 h. This confirmed that 4OI treatment reduced activation of the pro-apoptotic proteins: Caspase-9 (p-The125), Chk1 (p-Ser345), and Chk2 (p-Thr383) and enhanced levels of the anti-apoptotic protein: BCl2 (PSer70) (Fig. 6E). This 4OI treatment resulting in resistance to doxorubicin-mediated apoptosis correlated with lower mROS generation in both M-MDSCs (Fig. 6F) and PMN-MDSCs (Fig. 6G).

One proposed mechanism for itaconate's anti-inflammatory function is the regulation of nuclear factor erythroid 2–related factor 2 (Nrf2)[50]. Nrf2 is an important transcription factor serving as a sensor of oxidative stress. During oxidative stress, Nrf2 is released from the complex with regulatory protein Kelch-like ECH-associated protein 1 (KEAP1) to induce the transcription of many Nrf2-dependent genes. These gene products participate in the protection of the cell against oxidative stress-induced cell death[47,51].

To examine mechanisms of how itaconate decreases MDSC mROS and apoptosis, we generated MDSCs, pre-treated with 4OI (250 μM) and exposed them to doxorubicin (1 μM). Flowcytometry staining demonstrated that 4OI pre-treatment enhanced Nrf2 expression in M-MDSCs and PMN-MDSCs (Fig. 6H) within 4 h. 4OI treatment increased gene expression of the Nrf2 gene (*Nfe2l2*) (Fig. 6I). We wanted to confirm the role of Nrf2 in the regulation of MDSC antioxidation and apoptosis post β2-AR signaling. MDSCs were generated and treated with the Nrf2 inhibitor, ML385 (5 μM)[52] or DMSO control for one hour, then exposed to 4OI (250 μM) for 4 h, followed by doxorubicin (1000 ng/mL) for 20 h. Nrf2 functional blockade reversed the protective effect of 4OI against doxorubicin-induced apoptosis in M-MDSCs (Fig. 6J) and PMN-MDSCs (Fig. 6K) and decreased mROS production (Fig. 6L, M). We showed that 4OI treatment induced an increased antioxidant enzyme gene profile including *Prdx2, Gpx4, Sod2, Txnrd1, Gsr* after doxorubicin oxidative stress (Fig. 6N) which was reversed by ML385 pretreatment (Fig. 6O, P). These results confirm the key role of Nrf2 in the regulation of antioxidant machinery and MDSC survival. Antioxidant enzyme activity results in the generation of antioxidant molecules including glutathione (GSH). GSH is considered the most abundant endogenous antioxidant, neutralizing ROS products by peroxide oxidation via glutathione peroxidase[53,54]. Our scRNA-seq data showed an increased transcriptomic profile

correlating with GSH biosynthesis post β2-AR signaling (Fig. S8A). This was confirmed by higher GSH generation as determined by SIRM analysis (Fig. S8B, C). These results point to the role of itaconate (produced by STAT3 signaling after β2-AR activation) to activate antioxidant pathways, inducing resistance to doxorubicin-induced oxidative stress. However, these experiments used exogenous itaconate (4OI) treatment. There could be a difference in the function of endogenous itaconate versus 4OI that could be investigated in future studies.

## Targeting the STAT3 pathway, ATP and Itaconate metabolism along with doxorubicin modulates the anti-tumor response in an in vivo lymphoma model

Transcriptomic, proteomics and in vitro analyses indicated the key roles for the STAT3-pathway, ATP and itaconate metabolism in the regulation of MDSC metabolic fitness and survival. STAT3 signaling suppresses ATPIF1 expression leading to a shift of the ATP/AMP ratio. This causes downregulation of AMPK and activates the TCA cycle to generate itaconate, activating the antioxidant machinery via Nrf2. To confirm the importance of the STAT3-pathway, ATP and itaconate metabolism in the control of MDSC fitness and lymphoma treatment outcomes, we targeted the pathway in the EL4 (i.v.) lymphoma model. We had shown that blocking β-AR-induced STAT3 signaling in MDSCs by propranolol improved the efficacy of doxorubicin to increase survival (Fig. 1A). This phenotype correlated with MDSC function and was modulated primarily by β2-AR agonism and antagonism (Fig. 1K−M). To target downstream events, we administrated 4OI in the EL4 model with combination therapy (propranolol + doxorubicin). 4OI (50 mg/kg) treatment reversed the propranolol effect during combination treatment resulting in decreased survival (Fig. 7A). To confirm the MDSC role in the EL4 model, anti-Gr1 was given (twice weekly; i.p; 1 mg/kg), reversing the adverse effect of 4OI (Fig. 7B), highlighting deleterious effect of MDSCs on tumor control. Additionally, the effect of β2-AR signaling was evaluated in β2-AR−/− mice receiving the EL4 tumor. 4OI (50 mg/kg) injection of β2-AR−/− mice receiving doxorubicin reversed the beneficial effect of β2-AR blockade on survival (Fig. 7C). Pre-treatment in vivo of mice with 4OI (50 mg/kg) resulted in a reduced rate of MDSC apoptosis (Fig. 7D, E), reduced mROS production (Fig. 7F, G), and higher *Nrf2* expression (Fig. 7H−K) in response to doxorubicin treatment (20 mg/kg). We then blocked itaconate generation in combination with doxorubicin treatment using the Acod1−/− EL4 tumor mice model. Blockade of itaconate synthesis improved survival following doxorubicin treatment of EL4 tumor bearing mice (Fig. 7L), recapitulating the propranolol effect.

To examine the second part of the pathway (STAT3-induced ATP generation); we blocked the mitochondrial ETC. To assess the role of ATP generation in MDSC metabolic fitness post-STAT3 signaling, MDSCs were generated, followed by treatment with the ETC inhibitors rotenone, antimycin A, and metformin. We found that rotenone and

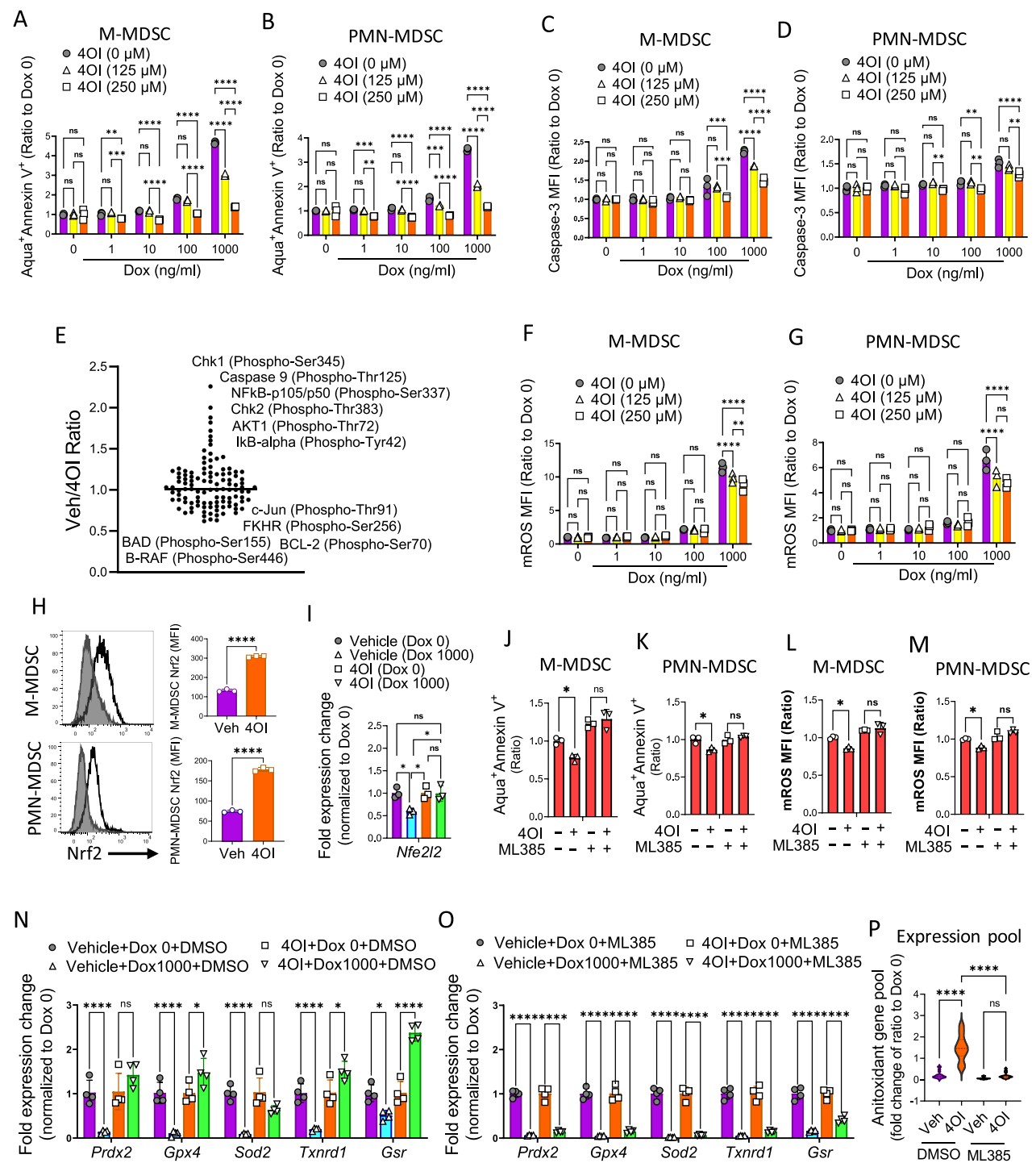

antimycin A caused significantly higher MDSC toxicity when compared to metformin (Fig. S9A). Therefore, we used metformin for subsequent experiments. We treated WT + ISO and β2-AR$^{-/-}$ control MDSCs with different metformin concentrations (0, 1, and 5 mM) and increasing doxorubicin concentrations (0, 1, 10 nM). Annexin V staining demonstrated that WT + ISO MDSCs were more resistant to apoptosis compared to control MDSCs (Fig. S9B). WT + ISO MDSCs also showed lower mROS production post exposure to doxorubicin and metformin compared to controls (Fig. S9C). Examination of ETC by seahorse analysis showed that metformin treatment reduced mitochondrial respiration (Fig. S9D) and ATP generation by MDSCs (Fig. S9E), indicating on-target effect of metformin on MDSC ETC. The gating strategy

to identify MDSCs subpopulations is shown in Fig. S10. We then utilized the EL4 (i.v.) lymphoma model; treating the mice with metformin (250 mg/kg) in drinking water or doxorubicin (day 7; i.v.; 4 mg/kg), or both. Metformin improved doxorubicin efficacy as measured by increased survival (Fig. 7M). To confirm the role of MDSCs in the EL4 model, anti-Gr1 (twice weekly; i.p.; 1 mg/kg) was given, resulting in improvement of metformin therapy alone to that of metformin+doxorubicin (Fig. 7M). We measured apoptosis during these metformin and doxorubicin therapy. The mice were pre-treated with Metformin (250 mg/kg) in drinking water for 7 days, followed by doxorubicin (i.v.; 20 mg/kg). After 18 h, increased apoptosis (Fig. 7N, O) and higher levels of activated caspase 3 (Fig. 7P, Q) were observed in MDSCs

**Fig. 6 | Itaconate drives the MDSC antioxidant machinery by regulation of the transcription factor Nrf2. A**–**D** MDSCs were generated in vitro as in Fig. 2. Cells were treated with 4-Octyl itaconate (4OI; 4 h), then doxorubicin (0–1000 ng/ml; 20 h). The apoptosis rate was measured in (**A**) CD11b⁺Ly6C⁺Ly6G⁻ cells (M-MDSC) and (**B**) CD11b⁺Ly6C⁺Ly6G⁺ cells (PMN-MDSC). Active caspase-3 levels were measured in (**C**) M-MDSCs and (**D**) PMN-MDSCs (**$p < 0.01$; ***$p < 0.001$; ****$p < 0.0001$). Two-way ANOVA with Tukey's multiple comparison tests. Data presented as mean ± SEM. Representative of two independent experiments with $n = 3$ technical replicates per group. **E** Phospho-proteomics analysis of apoptotic pathways by Apoptosis Phospho-Antibody Array on MDSCs treated with 4OI (250 μM; 4 h) then doxorubicin (100 nM; 20 h). Mitochondrial ROS (mROS) was measured in (**F**) M-MDSCs and (**G**) PMN-MDSCs treated as (**A**–**D**) (**$p < 0.01$; ****$p < 0.0001$). Two-way ANOVA with Tukey's multiple comparison tests. **H** Expression of Nrf2 protein assessed on MDSCs treated with vehicle or 250 μM of 4OI for 4 h. (****$p < 0.0001$). **I** Expression of *Nfe2l2* (Nrf2) mRNA in MDSCs treated with 4OI (250 μM; 4 h) then

doxorubicin (100 ng/ml; 20 h) (*$p < 0.05$). MDSCs were pre-treated with the Nrf1 inhibitor, ML385 (5 μM; 1 h), then 4OI (250 μM; 4 h), then doxorubicin (1000 ng/ml; 20 h). The apoptosis rate was measured on (**J**) M-MDSCs (*$p = 0.0322$) and (**K**) PMN-MDSCs (*$p = 0.0152$). mROS levels were measured in (**L**) M-MDSCs (*$p = 0.0118$) and (**M**) PMN-MDSCs (*$p = 0.0164$). One-way ANOVA with Bonferroni's multiple comparison tests. Data presented as mean ± SEM. Results in (**F**–**M**) are representative of two independent experiments with $n = 3$ technical replicates per group. (**N**, **O**) Cells treated as (**I**) were examined for expression of antioxidant genes (*$p < 0.05$; **$p < 0.01$; ***$p < 0.001$; ****$p < 0.0001$). Two-way ANOVA with Tukey's multiple comparison tests. Data presented as mean ± SEM. Representative of two independent experiments with $n = 4$ technical replicates per group. **P** Violin plots demonstrating pools of antioxidant genes presented in (**N**, **O**) (****$p < 0.0001$). One-way ANOVA with Bonferroni's multiple comparison tests. The raw data are provided in Source Data file.

isolated from the metformin plus doxorubicin therapy. These results confirm that blocking the STAT3-pathway, ATP and itaconate metabolism, by (1) blockade of β2-AR signaling, or (2) blocking ETC and ATP generation, or (3) blocking itaconate generation result in disrupted MDSC mitochondrial fitness determining the response to the doxorubicin chemotherapy anti-lymphoma effect and survival.

## Discussion

Metabolic reprogramming is a hallmark of cancer progression. It can support tumorigenesis by control of nutrient and oxygen availability, and the rewriting of metabolic processes that regulate the phenotype, function, and survival of immune cells[55]. MDSCs as key regulators of immune responses, rely on glucose, glutamine, and fatty acid oxidation to maintain metabolic fitness, critical to support their immunosuppressive functions[56]. Recently we reported that adrenergic stress-induced signaling of the β2-AR controls MDSC function[12,13] and metabolic capacity[57]. Extending these findings, we employed the β-AR signaling model for the induction of metabolic reprograming in MDSCs and examined the effects on the doxorubicin efficacy, a widely used chemotherapy agent. Our studies have uncovered the previously unknown role of ATP and Itaconate metabolism induced by β2-AR-STAT-3 pathway for controlling MDSC mitochondrial metabolic fitness, a determining factor for chemotherapy response.

We showed that combining propranolol with doxorubicin improved mice tumor model survival and the key role of MDSCs in the modulation of malignant cell growth was further confirmed by MDSC depletion by anti-Gr1 MDSC depleting antibody. These data are consistent with a role for MDSC β-AR signaling and tumor growth control. However, there could be additional effects of propranolol on the tumor cells and other immune cells[58–60] modulating overall tumor growth outcome. Targeting MDSCs was accomplished by anti-Gr1 antibody treatment in the β2-AR⁻/⁻ controls and by myeloid cell depletion using β2-AR⁻/⁻ LysM^Cre mice. A limitation in this model is that targeting β2-AR by LysM^Cre could potentially affect other myeloid cells including macrophages and conventional DCs.

Our study showed that β-AR signaling regulates MDSC mitochondrial function, mediated by STAT3 signaling after β-AR activation. MDSCs rely on STAT3 signaling, induced during differentiation[5,24]. This study confirms that activated STAT3 signaling induced by β2-AR signaling leads to enhancement of mitochondrial respiration and mitochondrial ATP generation. Additionally, we demonstrated that enhanced ATP generation was mediated by downregulation of ATPIF1 regulatory factor post-STAT3 signaling. Shifts in cell energy sources and ATP generation is detected by key energy sensors including AMPK[61]. AMPK pathway downregulation activates cellular antioxidant and anti-apoptotic mechanisms[62,63]. The regulation of AMPK is further responsible for shifts in energy pathways, including glycolysis, fatty acid oxidation and the TCA cycle[42,64].

Metabolic flux analysis using the Seahorse assay showed enhanced mitochondrial respiration following STAT3 signaling while the glycolysis rate was not affected, which was confirmed by SIRM with ¹³C₅-glutamine tracer. Investigation of the TCA cycle indicated that STAT3 signaling post-β-AR activation enhances glutamine metabolism via TCA cycle leading to the accumulation as citrate, cis-aconitate and itaconate. Itaconate has been reported to be a key regulator of immune responses post-metabolic reprogramming in mitochondria[47,65]. In macrophages, itaconate regulates several functions including metabolic reprogramming, control of immune responses and regulation of oxidative stress responses[47]. It also regulates tumor neutrophil survival affecting tumor progression[48]. During macrophage and neutrophil responses to oxidative stress, itaconate enhances expression of the transcription factor Nrf2 leading to activation of the cellular antioxidant machinery[47,48]. Previous studies have shown that itaconate modulates several activities including inhibition of succinate dehydrogenase (SDH) and glycolysis[47]. Our tracing studies did not show in MDSCs, a significant reduction in succinate post β-AR signaling suggesting itaconate accumulation. A recent study found no effect of accumulated itaconate on SDH within the tumor neutrophils[48]. Explanations for these contrasting findings could be due to the difference in the cell-type-specific regulation of TCA cycle, difference in the cell activation strategies, and different kinetics of itaconate generation[48].

One of the main functions of itaconate is through the regulation antioxidant genes[50,66]. We confirmed that high itaconate levels maintained MDSC resistance to doxorubicin mediated oxidative stress and intrinsic (mitochondrial) apoptosis by induction of Nrf2 and antioxidant pathways. This included activation of antioxidant enzymes and the generation of GSH.

Downregulation of STAT3 signaling by blocking β-AR signaling (a clinically relevant target), or disruption of STAT3-mediated itaconate result in an improved anti-tumor effect of doxorubicin. To target the ATP synthesis by oxidative phosphorylation, we blocked the ETC with metformin (a less toxic ETC inhibitor than rotenone or antimycin A)[67]. Metformin treatment improved doxorubicin efficacy. Interestingly, treatment with metformin alone was equivalent to metformin plus doxorubicin when MDSCs were depleted using anti-Gr1 antibody. Metformin plays a role in controlling MDSC ETC, thereby impeding the advancement of tumors. The impact of metformin on cancer management is currently under examination, with more than 400 studies registered on clinicaltrials.gov. It is crucial to delve deeper into understanding how metformin influences ETC regulation and other cellular functions. This includes exploring AMPK-dependent and AMPK-independent mechanisms, inhibiting mitochondrial respiration, inhibiting mitochondrial glycerophosphate dehydrogenase, and engaging mechanisms associated with the lysosome[68,69].

Some of the changes post β2-AR stimulation are subtle resulting from chronic, daily, physiologic stress which impairs the immune system. These effects can be cumulative, resulting in a significant

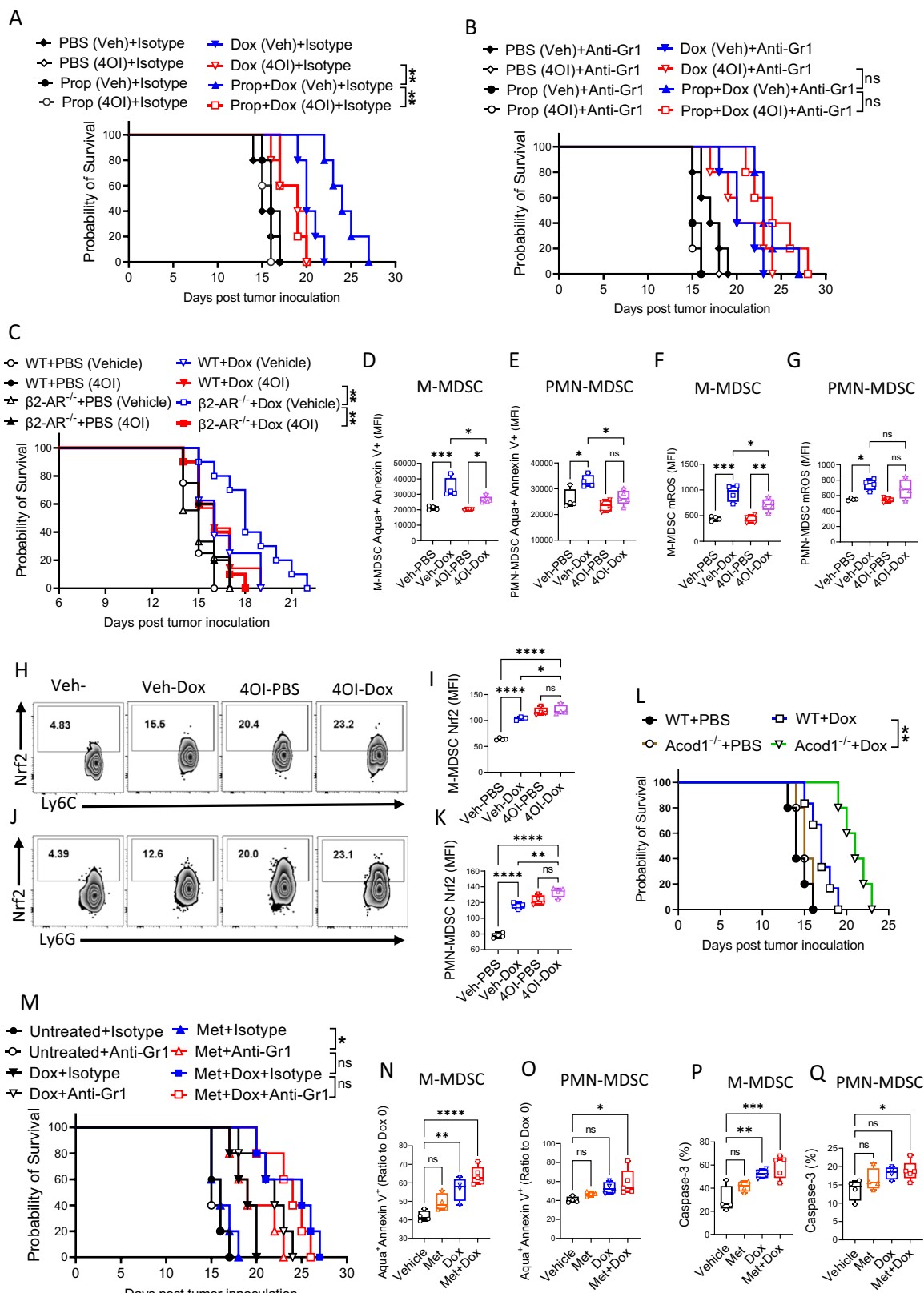

chronic, loss of tumor control. If major changes in immune function were seen daily, then experimental or clinical experimental or clinical subjects experiencing chronic stress would experience major, acute, life-threatening immune suppression. Chronic immunosuppression reduces the ability of doxorubicin to effectively kill tumor cells to control tumor growth.

Our study contains some biological and technical limitations. The application of β2-AR^fl/fl LysM^Cre transgenic mice models is limited by the pan myeloid cell expression of lysozyme which is not specific to MDSCs only[70]. Our studies show that ATPIF1 activation post STAT3 signaling is correlated with MDSC metabolic fitness. Further studies are needed to delineate all the mechanisms that lead to ATPIF1

**Fig. 7 | Itaconate and metformin can regulate MSDC metabolic fitness in vivo and determine lymphoma treatment outcome.** Survival of EL4 tumors bearing mice treated with 4OI (50 mg/kg) or vehicle every 3 days; Set (**A**) received isotype antibody and set (**B**) received anti-Gr1 (twice weekly; i.p.; 1 mg/kg). Each set received propranolol (daily; i.p.; 1 mg/kg), or doxorubicin (day 7; i.v.; 4 mg/kg), or both (**p = 0.0019; **p = 0.0027). Two-sided Mantel-Cox log rank test. **C** Survival of β2-AR$^{-/-}$ and WT mice bearing EL4 treated with 4OI (50 mg/kg) or vehicle every 3 days, then doxorubicin (day 7; i.v.; 4 mg/kg). Two-sided Mantel-Cox log rank test. **D**−**G** WT mice were pre-treated with 4OI (50 mg/kg) or vehicle for 4 h, then doxorubicin (i.v.; 20 mg/kg). After 18 h, the apoptosis rate was measured in (**D**) CD11b$^+$Ly6C$^+$Ly6G$^-$ (M-MDSC) and (**E**) CD11b$^+$Ly6C$^+$Ly6G$^+$ (PMN-MDSC). Mitochondrial ROS (mROS) was detected on (**F**) M-MDSCs, and (**G**) PMN-MDSCs (*p < 0.05; **p < 0.01; ***p < 0.001). One-way ANOVA with Bonferroni's multiple comparison tests. Data are presented as mean ± SEM. n = 4 mice per group. Nrf2 expression in (**H**, **I**) M-MDSCs and (**J**, **K**) PMN-MDSCs isolated from mice treated as in (**D**−**G**)

(*p < 0.05; **p < 0.01; ****p < 0.0001). One-way ANOVA with Bonferroni's multiple comparison tests. Data are presented as mean ± SEM. n = 4 mice per group. **L** Survival of EL4 bearing Aconitate Decarboxylase 1 deficient (Acod1$^{-/-}$) compared to WT mice treated with doxorubicin (day 7; i.v.; 4 mg/kg) (**p = 0.0031). Two-sided Mantel-Cox log rank test. **M** Survival of EL4 bearing mice receiving metformin (250 mg/kg; drinking water) or doxorubicin (day 7; i.v.; 4 mg/kg), or both. Set (**M**) received isotype control and set (**N**) received anti-Gr1 (twice weekly; i.p.; 1 mg/kg) (*p = 0.0180). Two-sided Mantel-Cox log rank test. WT mice pre-treated with metformin (250 mg/kg; drinking water; 7 days) received doxorubicin (i.v.; 20 mg/kg). After 18 h, the apoptosis rate was measured in by (**N**, **O**) Annexin V assay (*p = 0.0157; **p = 0.0024; ****p < 0.0001) and (**P**, **Q**) CaspGLOW™ Caspase-3 staining assay (*p = 0.0331; **p = 0.0055; ***p = 0.0005). One-way ANOVA with Bonferroni's multiple comparison tests. Data are presented as mean ± SEM. n = 4 mice per group. Source data containing independent experiments raw data are provided as a Source Data file.

expression and decreased ATP generation. While our experiments used an Acod1 knockout model or measured intracellular itaconate to study the role of the itaconate-NRF2 axis, several experiments relied on exogenous itaconate (4OI). The potential, functional difference between endogenous itaconate and 4OI treatment should be considered in the future studies.

Together, these studies reveal the STAT3, and ATP and Itaconate metabolism as key regulators of MDSC metabolic fitness which determines the response to doxorubicin therapy in the EL4 lymphoma model. STAT3 signaling downregulates ATPIF1 to shift the ATP/AMP ratio and suppress AMPK pathway. STAT3 signaling induces mitochondrial reprograming resulting in itaconate generation leading to activation of the antioxidant machinery via Nrf2 activation. These pathways counter doxorubicin-induced oxidative stress in MDSCs, resulting in higher MDSC survival and downregulation of antitumor immune responses. Blockade of these pathways led to improved outcomes in the EL4 lymphoma model. The STAT3 pathway ATP and Itaconate metabolism are targets for clinical trial development to improve current chemotherapy regimens in hematologic malignancies.

## Methods
### Mice and tumor cells
The C57BL/6J (B6 CD45.2$^+$), B6.129P2-Lyz2tm1(cre)Ifo/J (LysM$^{Cre}$), and C57BL/6NJ-Acod1em1(IMPC)J/J (Acod1) mice were purchased from Jackson Laboratories. β2-AR knockout (β2-AR$^{-/-}$ also known as *Adrb2$^{-/-}$*) mice on the C57BL/6 background were generated at Roswell Park Comprehensive Cancer Center (Roswell Park). All experiments were carried out using male and female mice that were 6−10 weeks of age. All mice were maintained in the Roswell Park Comparative Oncology Shared Resources (COSR) in specific pathogen-free conditions prior to use. The mice were maintained on a 12/12-h light/dark cycle, 22−26 °C, 50% humidity on average with sterile pellet food and water ad libitum. Animal work was done in accordance with the Institutional Animal Care and Use Committee (IACUC)-approved protocol according to Roswell Park animal care guidelines (protocol #1143 M). The EL4 T lymphoblast (TIB-39) cell line was purchased from American Type Culture Collection (ATCC) and tested and authenticated for mycoplasma before experimental use. EL4 cells were cultured in DMEM supplemented with 10% FBS, 1% L-glutamine, and 1% Penicillin/Streptomycin.

### Tumor model and treatment
To generate hematologic tumor models, $2 \times 10^6$ cells/mouse of EL4 (ATCC® TIB-39™) cells resuspended in 200 μL PBS were injected intravenously (i.v.) via mouse tail vein. For in vivo β-AR-blockade, propranolol (5−10 mg/kg) was injected daily intraperitoneally (i.p) in respective groups starting from day 1 until the endpoint. For doxorubicin treatments, doxorubicin (4 mg/kg) was injected (i.v.) once on day 7 post EL4 injection. For metformin treatments, 1.5 mg/ml of

metformin was dissolved in drinking water refreshed every 48 h. (to obtain 300 mg/kg dosage) in respective groups[67,71]. For 4-octyl itaconate (4-OI) treatments, the drug was dissolved in a PBS solution containing 40% 2-hydroxypropyl-b-cyclodextrin, and a dose of (50 mg/kg), vehicle control was injected every 3 days (i.p. in 200 μL) starting day 1 until the endpoint. For MDSC depletion, anti-Gr1 antibody (1 mg/kg) was injected i.p. twice weekly. The mice were monitored daily for posture, fur texture, and activity and euthanized if they showed signs of paralysis. For phenotyping experiments, the same dose of 4-OI (50 mg/kg in 200 μL i.p.) or vehicle was injected 4 h. before doxorubicin (20 mg/kg: i.v.) treatments, and cell compartments were examined at the 24 h. timepoint[72].

### In vitro MDSC generation
Bone marrow (BM) cells were harvested from the femur and tibia of the mice and red blood cells (RBCs) were removed using RBC lysis buffer. A total of $2 \times 10^6$ BM cells were cultured in 3 ml RPMI supplemented with 10% FBS, 1% L-glutamine, and 1% Penicillin/Streptomycin, in addition to granulocyte-macrophage colony-stimulating factor (rmGM-CSF) (40 ng/mL) and interleukin-6 (rmIL-6) (40 ng/mL) for 4 days. On day three, half of the media was gently replaced with fresh differentiation media. MDSCs (purity > 95% CD11b + ) were harvested on day 4 for subsequent experiments. For isoproterenol (ISO) treatment, 10 μM ISO was added to WT cell culture media starting day 0 and added fresh daily. Similarly, β2-AR$^{-/-}$ MDSC controls were generated with and without ISO treatment. To examine cell viability post electron chain transport (ECT) blockade in MDSCs, increasing concentrations of metformin hydrochloride: 0−5 mM (Cayman BioSystems), rotenone: 0−10 μM (Sigma), and antimycin A: 0−1 μM (Sigma) were added on day 3 of differentiation. Apoptosis was measured by the APC Annexin V apoptosis kit (Biolegends) as described below.

### Flow cytometry analysis
To obtain cells from the spleen or BM of mice, tissues were harvested and mechanically disintegrated, passed through a 70 μm strainer, followed by red blood cell (RBC) lysis. For in vitro cultured cells, MDSCs were harvested 4 days post differentiation. For flow cytometry assays, cells were suspended in FACS buffer (1% FBS in PBS) and stained for surface markers using anti-CD11b-BUV395 (M1/70), Ly-6C-BV421 (HK1.4), Ly-6G-APC or -PE (1A8), and Aqua LIVE/DEAD Fixable Dead Cell Stain dye. Surface staining was performed at 4 °C for 30 min in 200 μL staining volume. Intracellular staining was performed with the Fixation/Permeabilization Solution Kit (BD Bioscience) according to the manufacturer's protocol. After staining for surface markers, cells were fixed and permeabilized with fixation/permeabilization (BD PharMingen) and stained with Arg1-FITC (R&D: IC5868F) and iNOS2-PE (W16030C) in permeabilization buffer for 45 min at room temperature (RT). For the transcription factor Nrf2, isolated cells were stained intracellularly using the Foxp3/Transcription Factor Buffer Set

(eBiosciences) following the manufacturer's protocol. In this protocol, Nrf2-PE (D1Z9C) antibody incubation was performed for 45 min at RT. Cells were analyzed using the BD LSRFortessa™ Cell Analyzer (BD Bioscience), and analysis was performed using FlowJo V10 software. All antibodies are listed in the Key Resources Table.

## Annexin V staining

Apoptosis of MDSCs was measured using the APC Annexin V apoptosis kit (Biolegend). In brief, MDSCs for all experiments were generated and treated in vitro or isolated from mice. The cells were stained in FACS buffer (1% FCS in PBS) for surface MDSC markers (CD11b, Ly6C and Ly6G) and Aqua LIVE/DEAD Fixable Dead Cell Stain for 30 min at RT. Cells were washed with 1 ml of PBS and twice with 1X Binding Buffer (Biolegend) and resuspended in 200 μl of Binding Buffer. In each tube 5 μl of APC Annexin V was added and incubated for 15 min at RT. For the final step, 400 μl of Binding Buffer was added, and the cells were immediately analyzed using the BD LSRFortessa™ Cell Analyzer (BD Bioscience) and analysis was performed using FlowJo V10 software. Apoptosis rate was calculated based on the frequency of Aqua$^+$Annexin V$^+$ in each population (M-MDSC: CD11b$^+$ Ly-6C$^+$ Ly-6G$^-$ or PMN-MDSC: CD11b$^+$ Ly-6C$^-$ Ly6G$^+$) cells relative to untreated controls.

## Caspase 3 and 8 activities

Active caspase 3 and caspase 8 were measured by CaspGLOW™ Fluorescein Active Caspase-3 Staining Kit (ThermoFisher) and Casp-GLOW™ Fluorescein Active Caspase-8 Staining Kit (ThermoFisher), respectively. In these assays, cell permeable FITC-conjugated DEVD-FMK (for caspase 3) or FITC-conjugated IETD-FMK (for caspase 8), irreversibly binds to the active enzyme. For staining, $1 \times 10^6$ prepared cells in 300 μL staining buffer were incubated with MDSC surface marker antibodies and 1 μL of FITC-IETD-FMK or FITC-IETD-FMK for 1 h. at 37 °C. Cells were then washed twice with washing buffer and analyzed using the BD LSRFortessa™ Cell Analyzer (BD Bioscience).

## Mitochondrial ROS assay

To detect mROS, cells were prepared as mentioned above and incubated with MDSC surface marker antibodies in FACS buffer (PBS containing %2 FBS) for 30 min at 37 °C. Then 5 μM MitoSOX Red (ThermoFisher) was added for the last 10 min. Cells were analyzed using the BD LSRFortessa™ Cell Analyzer (BD Bioscience) after two washes with FACS buffer.

## scRNA-Seq and Real-time quantitative PCR

MDSCs were generated in vitro in presence of rmIL-6 and rmGM-CFS (40 ng/mL each) and 10 μM isoproterenol (ISO) as described above. On day 4, single cell libraries were generated using the 10X Genomics platform. First, Feature Barcode conjugated molecules were bound to cell surface proteins. Then live CD11b$^+$ cells were isolated using the SONY MA900 cell sorter (Sony Biotechnology). Single-cell suspensions were pooled together and assessed with AOPI stain using a Cellometer K2 automated cell counter (Nexcelom), to determine concentration, viability and the absence of clumps and debris that could interfere with single cell capture. Cells were loaded into the Chromium Controller (10X Genomics) where they were partitioned into nanoliter-scale Gel Beads-in-emulsion with a single barcode per cell. Reverse transcription was performed, and the resulting cDNA is amplified. Amplified cDNA is separated into full-length and Feature Barcode fractions using SPRI-Select beads (Beckman Coulter). The full-length amplified cDNA was used to generate libraries by enzymatic fragmentation, end-repair, a-tailing, adapter ligation, and PCR to add Illumina compatible sequencing adapters. Feature barcode derived cDNA was PCR amplified to incorporate Illumina adapter sequences and unique sample indexes. The resulting libraries were evaluated on D1000 screentape using a TapeStation 4200 (Agilent Technologies) and quantitated using Kapa Biosystems qPCR quantitation kit for Illumina. They were then pooled, denatured, and diluted to 300 pM with 1% PhiX control library added. The resulting pool was then loaded into the appropriate NovaSeq Reagent cartridge and sequenced on a NovaSeq6000 following the manufacturer's recommended protocol (Illumina Inc.).

Expression of selected genes were confirmed by real-time quantitative PCR using the ABI 7300 Real-Time PCR system (Applied Biosystems). Total RNA was extracted with RNeasy Mini Kit (QIAGEN) and RNA concentrations were determined using Nanodrop (Thermo Scientific). Total DNA-free RNA was used for mRNA isolation and library construction. For antioxidant enzymes, specific Taq-Man probes were used (ThermoFisher). After running the samples, expression of each gene was normalized to expression of the housekeeping gene (18S rRNA) as ΔCT = CT (gene X) – CT (18S rRNA). Then, alteration between groups were calculated as fold change = 2^(ΔΔCT). ΔΔCT is: CT (gene X) – CT (gene X at baseline). The specific TaqMan™ probes (Thermo-Fisher) used are listed in the Key Resources Table.

## Chromatin immunoprecipitation-Quantitative PCR (ChIP-qPCR)

$30 \times 10^6$ mice BM cells were harvested and treated with PBS or 10 μM isoproterenol (ISO) for 30 min. ChIP analysis was performed using the Pierce™ Magnetic ChIP kit (ThermoFisher) according to the manufacturer's instruction. The cells were crosslinked in 1% formaldehyde for 10 min at RT and neutralized with the provided glycine. Then cells were washed with PBS plus Halt Proteinase Inhibitor followed by incubation in membrane extraction buffer on ice. Extracted nuclei were digested with MNase for 15 min at 37 °C with mixing followed by sonication for 8 min (30 s ON/OFF cycles) using the Bioruptor™ UCD-200 water bath sonication device (Diagenode) on the "High" setting. Fragmented chromatin was equally distributed for immunoprecipitation. The fragmented chromatin was incubated with anti-pSTAT3 (Tyr705) (clone: D3A7) or anti-STAT3 (Polyclonal) antibodies overnight, at 4 °C while mixing. pSTAT3 or STAT3-bound chromatin was isolated with the provided Protein A/G Magnetic Beads by incubating for 2 h at 4 °C with mixing. Beads were washed and incubated at 65 °C for 40 min. Eluted chromatin was digested with Proteinase K for 1.5 h at 65 °C. Then, the DNA was purified using the provided spin columns prior to SYBR qPCR analysis. qPCR was performed using the SYBR™ Green PCR Master Mix (ThermoFisher) with the following cycling conditions: 95 °C for 10 min, 40x cycles of 95 °C for 15 s followed by 60 °C for 1 min. A Melt curve was generated with the following conditions: 95 for 15 s, 60 °C for 1 min, and continuous measurement until 95 °C. Primers used for qPCR are listed in Fig. S4.

## Bioenergetics analysis by Seahorse

Prepared MDSCs for different groups were seeded on Cell-Tak coated Seahorse XFe96 (Agilent) culture plates ($0.8-1 \times 10^5$ cells/well) in assay media. For oxygen consumption rates (OCR), assay media was DMEM supplemented with 1% BSA and 25 mM glucose, 1 mM pyruvate, and 2 mM glutamine. Basal OCR was taken for 30 min. To obtain maximal respiratory and control values, cells were stimulated with oligomycin (1 μM), FCCP (1.5 μM) and rotenone/antimycin A (0.5 μM). In designated experiments, to confirm STAT3 signaling post β-AR activation, MDSCs were generated in presence of ISO and increasing concentration of JSI-124 (0–3 μM) were added to the cells and used in the Seahorse assay.

## Tracer studies in cell cultures

For tracing experiments, MDSC + ISO and MDSC + PBS in one set and β2-AR$^{-/-}$ + ISO and β2-AR$^{-/-}$ + PBS in another set were generated using rmIL-6 and rmGM-CSF (40 ng/mL each) for 4 days. On day 4, the cells were incubated with tracing media: Gln-free RPMI medium supplemented with 10% dialyzed FBS (Life Technologies), 20 mM HEPES, 0.05 mM 2-mercaptoethanol, and 1% penicillin-streptomycin plus 2 mM $^{13}C_5$-glutamine ($^{13}C_5$-Gln) (Cambridge Isotope Laboratories,). At 2 h, 8 h. and 24 h. incubation timepoints, the cells were washed twice

with PBS, once quickly with distilled water, and quenched with 60% cold acetonitrile in water. Polar metabolites were extracted by the solvent partitioning method with a final $CH_3CN:H_2O:CHCl_3$ (2:1.5:1, v/v) ratio, followed by a second extraction including methanol, and lyophilized as previously described[73,74].

## IC-UHR-MS analysis
We examined the polar extracts by Ion Chromatography-UltraHigh Resolution Mass Spectrometry (IC-UHR-MS). The samples were analyzed by Dionex ICS-5000+ ion chromatography interfaced to a Thermo Fusion Orbitrap Tribrid mass spectrometer (Thermo Fisher Scientific). Samples were run in the detailed steps of IC-MS as described previously[75]. IonPac AS11-HC-4 µm RFIC&HPIC (2 × 250 mm) column and an IonPac AG11-HC-4 µm guard column (2 × 50 mm) were used for Ion chromatography. The flow rate of column was set at 0.38 mL/min and a temperature of 35 °C. To aid vaporization in the heated electrospray ionization (HESI) unit, methanol (0.06 mL/min) was added post-column as a makeup solvent. The HESI vaporizer temperature was set at 400 °C with the sheath gas set at 35 arbitrary units (~3 L/min) and auxiliary nitrogen flow at 4 arbitrary units (~4 L/min). The KOH gradient was generated in the IC as, 8 min of 1 mM KOH for the initial equilibration, then 1 mM KOH for 2 min, followed by 10 µL of sample injection. This was followed by linear ramp to 40 mM KOH from 2 to 25 min, a second linear ramp to 100 mM from 25 to 39 min, then held at 100 mM up to 50 min. The reset program to run the next sample was set by ramping down to 1 mM KOH at 50.1 min and held until 52.5 min. All data were collected as MS1 under this UHR condition. We integrated the peak areas using the Thermo TraceFinder (version 3.3) software package. The peak areas were normalized for natural abundance as previously described[73]. The concentration of each metabolite was normalized based on protein concentrations calculated from peak areas using the MS response factor for authentication standards of known concentration analyzed before and after the sample analysis (reposted as µmole/g protein)[75].

## Western blot
MDSCs in different groups were generated as described above. After the described treatment times, cells were centrifuged, washed with PBS, and frozen at −80 °C. Cells were lysed using RIPA Lysis and Extraction Buffer (ThermoFisher) supplemented with a Pierce protease and phosphatase inhibitor mini tablet (ThermoFisher) and 0.1 M PMSF (ThermoFisher). The Pierce BCA protein assay (ThermoFisher) was used to quantify protein concentration in sample lysates using a clear, flat-bottom, 96-well plate, on a plate reader (Synergy H1). SDS-PAGE was used for resolution of proteins, which were then transferred to a polyvinylidene difluoride membrane (Millipore) and blocked with 5% nonfat milk or 5% BSA (ThermoFisher Scientific) in Tris buffered saline (Bio-Rad) with Tween 20 (Bio-Rad) in accordance with primary antibody incubation specifications. All membranes were incubated overnight, at concentrations of 1:1000 for ATPIF1 (5E2D7; ThemoFisher), 1:1000 for AMPK (Cat# 5831, Cell Signaling), 1:1000 for pAMPK-172 (Cat# 2535; Cell Signaling) and 1:1000 for Vinculin (Cat# 13901, Cell Signaling). Anti-rabbit (Cat# 7074, Cell Signaling) and anti-mouse (Cat# 7076, Cell Signaling) horseradish peroxidase–conjugate secondary antibodies were used at a concentration of 1:3000. Membranes were developed with ECL-substrate (Bio-Rad) and images were captured using the LI-COR Odyssey Fc (Licor) in colorimetric mode (to visualize the marker) and chemiluminescence mode (to visualize target proteins).

## Protein phosphorylation profiling
Qualitative protein phosphorylation profiling for apoptotic pathways was done using the Apoptosis Phospho-Antibody Array kit (Full Moon BioSystems) according to manufacturer's protocol. The assay is a high-throughput ELISA based antibody array using 247 site-specific and serine/threonine and tyrosine phospho-specific antibodies. In our study, MDSCs were generated in presence of rmIL-6 and rmGM-CSF (40 ng/ml each) for 4 days and β-AR -signaling was stimulated by ISO treatment (10 µM). Cells were washed twice with PBS and lysed by lysis buffer containing protease and phosphatase inhibitor mini tablets (ThermoFisher). Extracted proteins were biotinylated and incubated on array antibodies coated on antibody slides. After completing wash steps, detection was done by Cy3-labled streptavidin. Slides were scanned and for each spot on the array, median signal intensity (MSI) was extracted from array image. Average signal intensity of target replicates was calculated, value for each target was normalized using background and internal controls (beta-actin, GAPDH and negative controls) and then fold change (FC) between groups was calculated. The phosphorylation levels of following markers were used to determine Jak/STAT pathway activity: pSTAT3 (Tyr705), pSTAT2 (Tyr690), pSTAT5B (Ser731), pSTAT6 (Thr645), pJAK2 (Tyr1007), pJAK2 (Tyr221), pAKT1 (Ser473), pAKT1S1 (Thr246), pAKT2 (Ser474), pRaf1 (Ser259). The phosphorylation levels of following markers were used to determine AMPK pathway activity: pAMPKbeta1 (Ser182), pMAP3K7/TAK1 (Thr184), p4E-BP1 (Thr45), pACC1 (Ser80), pP70S6K (Thr421), pP70S6K (Ser424), pPDK1 (Ser241), pCyclin D1 (Thr286), pEEF2 (Thr56), peIF4E (Ser209), pPP2A-alpha (Tyr307), pFOXO1/3/4-pan (Thr24/32).

## Statistics and reproducibility
Sample sizes were chosen based on previous publications and were sufficient for statistical analysis. No data were excluded from analysis. Animals were randomly assigned to each group. Randomization was not required for biochemical and in vitro experiments. Cellular and biochemical experiments were not blinded. All the statistical analysis were done using GraphPad Prism 6 (GraphPad) and Excel software. Differences between groups were analyzed using the Student $t$ test (for two groups) or one way ANOVA (for multiple groups), followed by the post-hoc Tukey test. Animal survival (Kaplan−Meier survival curves) was analyzed by log-rank test. Differences between groups were rated significant at values of $p < 0.05$. In the figures all data are shown as mean ± SEM and $*p \leq 0.05$, $**p \leq 0.01$, $***p \leq 0.001$, $****p \leq 0.0001$, ns = not significant.

## Reporting summary
Further information on research design is available in the Nature Portfolio Reporting Summary linked to this article.

## Data availability
The raw and processed sequencing data (single cell RNA-seq) generated in this study have been deposited in the Gene Expression Omnibus (GEO) database under accession number GSE254512. All data supporting the findings of this study are available within the article or in the Supplementary Information. Source data containing independent experiments raw data are provided with this paper. Source data are provided with this paper.

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

## Acknowledgements

The authors thank Jeanne M. Prendergast, Feng Li, Eugene Kononov, Courtney C. Ryan, and Suzanne M. Hess, for technical assistance and support, and the Genomics Shared Resource, Translational Imaging Shared Resource, Comparative Oncology Shared Resources (COSR), and the Flow Cytometry Core Facility for expert support. MS were recorded using the Metabolism Shared Resources at the University of Kentucky supported in part by P30CA177558 (to B.M. Evers). This project is supported by K99 HL155792, R00 HL155792, and Roswell Park Alliance Foundation (to H.M.), R01 CA205246 (to E.R.), F30CA265127 (to C.R.M.), and a donation from Brendan and Elise McCarthy (P.L.M.), and NCI grant P30CA016056.

## Author contributions

Experimental design and execution were conducted by S.D., C.R.M., M.P., H.M. and J.E.C.; M.G., Q. Y. and N.R. helped with in vitro and in vivo experiments. Data interpretation was performed by S.D., P.L.M., E.A.R., H.M. scRNA-seq transcriptomics and bioinformatical analysis were done by P.K.S. and J.W. Tracing studies were performed by S.D., R.M.H., A.N.L., T.W.F. The manuscript was written and edited by S.D., C.R.M., P.L.M., E.A.R., H.M.

## Competing interests

P.L.M. has received honoraria from and participated in advisory boards from Bristol Myers Squibb, Bluebird, Celgene, Janssen, Juno, Karyopharm, Magenta Therapeutics, Oncopeptides and Takeda. The other authors have no competing interests to declare.
