## [Peer Review File · Nature Communications]

Myeloid-derived suppressor cell mitochondrial fitness governs
chemotherapeutic efficacy in hematologic malignanciesReviewers' Comments:

Reviewer #1 - MDSC, immunometabolism (Remarks to the Author):

Saeed Daneshmandi and co-authors described the ATP/Itaconate pathway as a critical regulator of mitochondrial metabolic fitness in MDSCs and response to doxorubicin chemotherapy in the model of EL4 lymphoma. Authors showed that triggering STAT3 signaling via activation of adrenergic receptor caused metabolic reprogramming of MDSCs, manifested by OXPHOS and ATP generation. These effects were driven by a reduction in ATPase inhibitory factor 1 (ATPIF1) expression. STAT3 signaling in MDSCs activated a metabolic shift characterized by enhanced glutamine consumption via the TCA cycle. Metabolized glutamine accumulated as itaconate which downregulates mitochondrial ROS via regulation of Nrf2. Targeting the STAT3 induced ATP/Itaconate pathway by blocking AR signaling, the electron transport chain and ATP generation, or itaconate generation, resulted in disrupted mitochondrial fitness of MDSCs, which increased the response to doxorubicin and slowed the progression of lymphoma.

Authors presented quite extensive assessment of metabolic pathway downstream of AR. Most of the aspects of the study recapitulated known facts about regulation of mitochondrial function by STAT3 and contribution of itaconate. However, authors built an axis linking all elements of the pathway together and implicated AR as starting point of this axis. The involvement of this pathway in response to doxorubicin is also very interesting observation. Therefore, to the opinion of this reviewer, this study is sufficiently novel and biologically relevant.

However, the main concerns appeared during the analysis of the data. I listed the concerns in the order of importance

1. In all experiments started from Figure 2 authors did not use the most obvious control - WT cells/mice. As a control they used b2AR^{-/-} cells. So they compare cells treated with b2AR agonist with cells lacking the receptor. The differences are clear, albeit rather modest. It is understandable to start with this approach to better discriminate between two polarized conditions. However, then one would expect to see the differences between agonistic approach and wild-type cells. Would be authors concerned with biological relevance of their data? What happens if differences were not significant. Would it question the conclusions? It is simply unclear how relevant this effect to real biological situation.
2. The data in Figure 7 are not convincing. It is not clear why well known effect of metformin in vivo was associated specifically with MDSC in this system. It could be totally independent. Experiments with 4-OI suffer from the same limitations and again with absence of appropriate control WT + DOX.
3. The fact that itaconate would drive expression of NRF2 as compensatory mechanism is known. Authors claimed that NRF2 is responsible for the observed biological effect. However, no evidence were provided to support this claim. At minimum inhibition of NRF2 needs to be performed to assess the downstream consequences.

4. Figure 5F-I. The differences are extremely small. In apoptosis measurement (F,G) one would expect stronger effect to be confident in biological effect. In figure H and I, this is MFI. Such level of differences (10-15%) is difficult to be considered as meaningful even if authors showed statistical significance.

5. In figure 3A, why glycolysis (acidification rate) was not measured? Why did the data presented only at gene expression level? Monocytes and PMN are known to have different level of OXPHOS. Would it be important to assess them differently?

6. In Figure 1, authors demonstrated survival benefits of combination of propranolol with doxorubicin and then demonstrated changes in MDSC. They build concept of priming of MDSC changes in these effects. However, the obvious explanation of those changes would be that MDSC were changed because differences in tumor growth. This aspect was not clearly addressed. The only experiment that suggest this link is Figure 1K where authors used LysMcre system to delete AR. There are two issues there. First, LysMcre is expressed on most macrophages and many DC and thus requires changes in interpretation. Second, authors did not assess the changes in MDSC in these experiments.

7. Figure 3K - without clear background, this WB looks rather artificial.

Reviewer #2 - Metabolism, mitochondria (Remarks to the Author):

In this study, Daneshmandi et al investigated the role of β -adrenergic signalling in regulating myeloid derived suppressor cell (MDSC) function during haematological malignancy. Using genetic and pharmacological models, they show improved survival when combining the chemotherapeutic doxorubicin with propranolol (β -AR blocker) and decreased MDSC populations, which was recapitulated with β 2-AR $^{-/-}$ (global and myeloid specific). Mechanistically, the loss of MDSCs is attributed to increased apoptosis. β -AR activation was found to increase mitochondrial respiration in a STAT3 dependent manner, which is suggested that this is mediated by repression of the ATP synthase inhibitor ATP1F1. It's suggested that β -AR activation supports itaconate production to activate NRF2 and promote an antioxidant response. Both events are suggested to limit MDSC apoptosis and so blocking β -AR synergises with doxorubicin to improve tumour control.

Strengths

1. Opening in vivo work appears convincing and interesting.

Weaknesses

1. There appear to be a lack of important controls in many of the in vitro experiments. Notably just comparing WT+ISO to β 2-AR $^{-/-}$ MDSCs is not sufficient. The comparisons should have included both WT MDSCs (with no ISO) and β 2-AR $^{-/-}$ with ISO. It is very hard to draw conclusions without appropriate controls from these experiments. This is critical as all the metabolic work and other mechanistic experiments were performed without these controls and it is not clear why these controls were not

performed, and the data becomes difficult to interpret as a result.

2. ATP1F1 link is correlative.

3. Itaconate is a well-established inhibitor of succinate dehydrogenase (SDH), which suppresses respiration. A concomitant increase in mitochondrial respiration and itaconate is difficult to understand and interpret in this context, again this is muddled by the experiment design. The combination of STAT3/ATP-itaconate doesn't represent an axis and this is unusual to suggest.

4. The tracing approach is also quite over complicated, not super informative especially because the lack of controls. It would have been informative to assess levels of these metabolites with unlabelled metabolomics (with the appropriate controls) and follow up with single-labelled tracers (U-13C-glutamine for e.g.) to assess isotopologue distributions and differences in glutamine carbon incorporation into the TCA cycle.

5. Acod1^{-/-} in vitro data is weak/unconvincing so much of the itaconate angle relies on the itaconate ester 4-octyl itaconate (4-OI), which is a more potent NRF2 activator.

Reviewer #3 - sympatic nervous system (B-adrenergic), cancer

[Editorial Note: This reviewer did not leave any comments to the authors]

Reviewer #4 - Blood cancer, metabolism, some sequencing (Remarks to the Author):

This manuscript by Daneshmandi et al proposes that beta-adrenergic signaling in myeloid derived suppressor cells (MDSCs) induces a series of metabolic effects that make the MDSCs more resistant to doxorubicin and thereby impairs survival in hematologic malignancies. Specifically, the manuscript proposes that adrenergic receptor signaling activates STAT3 which suppresses ATP1F1 leading to enhanced OXPHOS and ATP generation resulting in a higher ATP/AMP ratio leading to inhibition of AMPK activity and increased glutamine-derived itaconate production. Itaconate in turn buffers ROS via NRF2 activation and makes MDSCs more resistant to doxorubicin (requires ROS for MOA). The manuscript combines in vitro studies of MDSCs with in vivo models of lymphoma (EL4). The manuscript is generally well written and the data are clearly presented. However, the improper use of controls substantially weakens the conclusions (see below). In addition, most of the mechanistic connections are circumstantial and indirectly demonstrated.

Major:

1) The conclusions are substantially weakened by the lack of proper controls. Because of this, most of the data are inconclusive as presented. Throughout the manuscript, various genetic mutants are used as "controls" for wildtype cells treated with drugs. Most notably WT + isoproterenol MDSCs are repeatedly compared to B2AR KO MDSCs without isoproterenol (Figs 2, 3, 4, 5). It is not clear why this comparison was chosen, and it seems likely that it will magnify most of the phenotypes. Importantly, the manuscript does not demonstrate that B2AR KO MDSCs develop or function normally. More appropriate comparisons would be (WT + vehicle) versus (WT + isoproterenol) versus (B2AR KO + vehicle) versus

(B2AR KO + isoproterenol). Similar issue with Acod1 KO mice; for Fig 7K, what does survival look like in Acod1 KO + PBS?

Other:

2) For all the in vivo experiments, it is not demonstrated (only inferred) that the effects on lymphoma survival are due to loss of MDSCs. It would help to show effect on survival with antibody or diphtheria toxin mediated depletion of myeloid cells, and further demonstrate that the various interventions have no effect on survival when MDSCs are not present. The exception to this is the experiments in Fig 1 with LysM Cre mediated deletion of B2AR which better narrow the effect to MDSCs.

3) Metformin has many proposed mechanisms of action beyond complex I inhibition. For Fig 7, it should be demonstrated that metformin is inhibiting ETC function at the doses used. Likewise, effects of metformin on the relevant metabolic parameters should be shown (e.g. itaconate levels, ROS, etc) to better support the notion that it is acting through the proposed mechanism.

4) The single cell RNA sequencing data in Fig 2H and 3C is very difficult to decipher when presented in this way.

5) For Fig 3E, it is not clear how this measure of JAK-STAT activation was performed. Should provide more details and/or show Western blot as well.

6) For Fig 3 experiments with JSI-124, it is not demonstrated that this is drug acting on target. Should be shown by western blot or better with genetic experiment targeting STAT3.

7) For Fig 3, the Atpif1 data is correlative and this should be more explicitly acknowledged in the text.

Reviewers' comments:

Reviewer #1 - MDSC, immunometabolism (Remarks to the Author):

Comment:

Authors presented quite extensive assessment of metabolic pathway downstream of AR. Most of the aspects of the study recapitulated known facts about regulation of mitochondrial function by STAT3 and contribution of itaconate. However, authors built an axis linking all elements of the pathway together and implicated AR as starting point of this axis. The involvement of this pathway in response to doxorubicin is also very interesting observation. Therefore, to the opinion of this reviewer, this study is sufficiently novel and biologically relevant.

However, the main concerns appeared during the analysis of the data. I listed the concerns in the order of importance.

Comment 1: In all experiments started from Figure 2 authors did not use the most obvious control - WT cells/mice. As a control they used $\beta 2AR^{-/-}$ cells. So they compare cells treated with $\beta 2AR$ agonist with cells lacking the receptor. The differences are clear, albeit rather modest. It is understandable to start with this approach to better discriminate between two polarized conditions. However, then one would expect to see the differences between agonistic approach and wild-type cells. Would be authors concerned with biological relevance of their data? What happens if differences were not significant. Would it question the conclusions? It is simply unclear how relevant this effect to real biological situation.

Response: *We thank you for the comments regarding the biologic relevance of the data. In response to the concerns about the need for additional controls, we repeated all in vitro experiments with more extensive control groups (WT + vehicle) vs (WT + isoproterenol) vs ($\beta 2AR$ KO + vehicle) vs ($\beta 2AR$ KO + isoproterenol). These experiments are now presented in Figures 2-5 and supplementary Figures S2, S4, S6-S9. We compared WT β -AR activation with isoproterenol (ISO)) with the (WT + vehicle) along with ($\beta 2AR$ KO + vehicle) vs ($\beta 2AR$ KO + ISO) to control for $\beta 2AR$ signaling specificity. The new experiments more strongly support the role of the ATP/Itaconate pathway post- $\beta 2AR$ signaling in the control of MDSC mitochondrial fitness following doxorubicin treatment and tumor control in the EL4 lymphoma model.*

Comment 2: The data in Figure 7 are not convincing. It is not clear why well-known effect of metformin *in vivo* was associated specifically with MDSC in this system. It could be totally independent. Experiments with 4-OI suffer from the same limitations and again with absence of appropriate control WT + DOX.

Response: *Thank you for identifying this issue. In response, we conducted further experiments to delineate the role of MDSCs in the anti-tumor effects of metformin and the role of 4OI treatment in vivo. As noted in the Response to Comment 1, we repeated the in vivo experiments using anti-Gr1 MDSC depleting antibody. These are added to figure 7 (Figure 7A-B and 7M). The*

administration of the anti-Gr1 MDSC depleting antibody blocked the suppressive function of 4OI treatment (Figure 7A-B; page 17, lines 409-414). MDSC depletion by the anti-Gr1 antibody enhanced the metformin treatment effect, comparable to metformin+doxorubicin therapy (Figure 7M; page 18, lines 435-438). These results suggest an important role for metformin in regulating the MDSC Electron transport chain (ETC) and tumor response. We have added to the Discussion the potential effects of metformin on tumor cells and other cell functions beyond regulation of ETC which would be investigated in future studies (pages 20-21, lines 502-511).

Comment 3: The fact that itaconate would drive expression of NRF2 as compensatory mechanism is known. Authors claimed that NRF2 is responsible for the observed biological effect. However, no evidence were provided to support this claim. At minimum inhibition of NRF2 needs to be performed to assess the downstream consequences.

Response: *We fully agree and conducted a new experiment using a NRF2 inhibitor (ML 385) to examine the effect of 4OI on NRF2 functions (Apoptosis rate, mROS, antioxidant gene expression). These experiments are in Figure 6 (Page 15 and 16, lines 380-389). Inhibition of NRF2 function by ML 385 reversed the protective effect of NRF2 expression post β 2-AR signaling, confirming the role of NRF2 in the MDSC response to doxorubicin (Dox) treatment.*

Comment 4: Figure 5F-I. The differences are extremely small. In apoptosis measurement (F,G) one would expect stronger effect to be confident in biological effect. In figure H and I, this is MFI. Such level of differences (10-15%) is difficult to considered as meaningful even if authors showed statistical significance.

Response: *This is an insightful comment and addresses an important issue that we discuss often.. (Please also see our response regarding these findings which brought up by Reviewer 2, Comment 6) While statistically significant, we agree that some of the results can be considered subtle. But we believe it is important to recognize that the stress in these experiments would be reflective of mild but chronic physiologic stress where the immune system is impaired to a mild extent each day. While subtle, these physiologically relevant effects can be cumulative, and would be detrimental to long term tumor control. In fact, we hypothesize that if greater effects were seen with chronic stress, then animals (including humans) would experience major, acute, life-threatening immune suppression daily. Although the effects are subtle, they result in a statistically significant enhancement of immunosuppression, promoting tumor growth. Daily, chronic immunosuppression reduces the ability of doxorubicin to kill tumor cells more effectively. We have better addressed this issue in the Discussion (page 21, lines 513-518). We also re-analyzed the data in Figures 5H and I. We added more controls, presenting the data as Change in the ratio to Dox 0 (Figure 5H, page 14, lines 346-347).*

Comment 5: In figure 3A, why glycolysis (acidification rate) was not measured? Why did the data presented only at gene expression level? Monocytes and PMN are known to have different level of OXPHOS. Would it be important to assess them differently?

Response: *Thank you for the comment and recommendation. The glycolysis (acidification rate) and metabolite concentrations measured by tracing data are now presented in the new Supplementary Figure 7 (S7). The glycolytic pathway activity was not changed significantly by β 2-AR signaling in MDSCs (Pages 12 and 13, lines 303-314). We sorted M-MDSC and PMN-MDSCs separately to determine if β 2-AR signaling on MDSC metabolism is different between M-MDSC and PMN-MDSC (new supplementary Figure S4). M-MDSCs demonstrate higher rate of OXPHOS than PMN-MDSCs (Figure S4B) with both M-MDSCs and PMN-MDSCs having similar OXPHOS enhancement by β 2-AR stimulation (Figure S4A-B) (page 8, lines 182-185).*

Comment 6: In Figure 1, authors demonstrated survival benefits of combination of propranolol with doxorubicin and then demonstrated changes in MDSC. They build concept of primacy of MDSC changes in these effects. However, the obvious explanation of those changes would be that MDSC were changed because differences in tumor growth. This aspect was not clearly addressed. The only experiment that suggest this link is Figure 1K where authors used LysMcre system to delete AR. There are two issues there. First, LysMcre is expressed on most macrophages and many DC and thus requires changes in interpretation. Second, authors did not assess the changes in MDSC in these experiments.

Response: *Thank you for pointing this out. We have repeated the in vivo experiments in Figure 1A (now 1A-B), Figure 7A (now 7A-B), and Figure 7L (now 7M) using anti-Gr1 MDSC depleting antibody to confirm the role of MDSCs during doxorubicin treatment. The new data confirm the effect of doxorubicin on MDSCs. We show that the ATP/Itaconate pathway participates in the control of MDSC mitochondrial fitness during doxorubicin therapy and the subsequent tumor response. We have added text to discuss possible direct and indirect drug effects on tumor burden and the MDSC population (pages 19-20, lines 481-495 and pages 20-21, lines 502-518). We also discuss that LysM^{Cre} might affect other myeloid cells including macrophages and conventional DCs (page 19, lines 462-468) as follows:*

“These data are consistent with a role for MDSC β -AR signaling and tumor growth control. However, there could be additional effects of propranolol on the tumor cells and other immune cells (58-59-60) modulating overall tumor growth outcome. Targeting MDSCs was accomplished by anti-Gr1 antibody treatment in the β 2-AR^{-/-} controls and by myeloid cell depletion using β 2-AR^{-/-} LysM^{Cre} mice. A limitation in this model is that targeting β 2-AR by LysM^{Cre} could potentially affect other myeloid cells including macrophages and conventional DCs”.

Comment 7: Figure 3K - without clear background, this WB looks rather artificial.

Response: *Thank you for the comment. We repeated the western blot data regarding ATP1F1 expression (now Figure 3I) using a traditional protocol to confirm the results we obtained using Jess automatic western blot (page 9, lines 218-219).*

Reviewer #2 - Metabolism, mitochondria (Remarks to the Author):

Strengths

Comment 1: Opening in vivo work appears convincing and interesting.

Response: *Thank you for this comment regarding the in vivo data.*

Weaknesses

Comment 2: There appear to be a lack of important controls in many of the in vitro experiments. Notably just comparing WT+ISO to b2-AR^{-/-} MDSCs is not sufficient. The comparisons should have included both WT MDSCs (with no ISO) and b2-AR^{-/-} with ISO. It is very hard to draw conclusions without appropriate controls from these experiments. This is critical as all the metabolic work and other mechanistic experiments were performed without these controls and it is not clear why these controls were not performed, and the data becomes difficult to interpret as a result.

Response: *We fully agree and thank the reviewer for pointing this out. (Please also see response to Reviewer 1, Comment 1). We initially reported the difference between WT and $\beta 2AR^{-/-}$ cells in vitro, to recapitulate and compare groups from the in vivo experiments (i.e., low vs. high β -AR signaling). For the revised manuscript, we have repeated the in vitro experiments adding more extensive control groups (WT + vehicle) vs (WT + ISO) vs ($\beta 2$ -AR KO + vehicle) vs ($\beta 2$ -AR KO + ISO). These are presented in Figures 2-5 and supplementary Figures S2, S4, S6-S9. In these experiments, we compared the β -AR activation (ISO) with WT to examine the effect of β -AR signaling and used ($\beta 2$ -AR KO + vehicle) vs ($\beta 2$ -AR KO + ISO) to control the specificity of the $\beta 2$ -AR signaling. The new experiments now more strongly support our original hypothesis regarding the role of the ATP/Itaconate pathway post- $\beta 2$ -AR signaling in the control of MDSC mitochondrial fitness following doxorubicin treatment and tumor control in the EL4 lymphoma model.*

Comment 3: ATP1F1 link is correlative.

Response: *To address this valid concern, we employed the specific STAT3 inhibitor (JSI-124) to confirm function of STAT3 signaling on Atp1f1 expression (Fig. 3K). Additionally, we performed ChIP-qPCR to confirm that STAT3/pSTAT3 binding to Atp1f1 promoter post AR signaling regulates the Atp1f1 expression (Fig. 3L). Unphosphorylated STAT3 (STAT3) binds to the Atp1f1 promoter as a regulator of Atp1f1 expression in untreated MDSCs. $\beta 2$ -AR stimulation enhances STAT3 phosphorylation (pSTAT3) resulting in a reduced STAT3/pSTAT3 ratio (Fig. 3E-F). This leads to the lower availability of STAT3 to bind to the Atp1f1 promoter causing a reduction in Atp1f1 expression (Fig. 3L). Blockade of STAT3 phosphorylation by JSI-124 reverses this effect, leading to enhanced ATP1F1 expression (Fig. 3K). The new data are presented in Figure 3E-L. The following text has been added to the manuscript (pages 9-10, lines 221-242):*

*“STAT3 signaling is important for shaping MDSC differentiation and function. This includes regulating the gene expression profile toward a suppressive phenotype and functions. Although pSTAT3 regulates gene expression at early time points post activation, unphosphorylated STAT3 is responsible for pSTAT3 non-responding genes at later time points (PMID: 17510282). Unphosphorylated STAT3 accumulates in response to gp130-linked cytokines (e.g. IL-6), activating transcription. Unphosphorylated STAT3 sustains cytokine-dependent signaling at late times through a distinct mechanism from pSTAT3. To examine how the STAT3 pathway post β -AR signaling reduces *Atpif1* expression (needed for MDSC-enhanced OXPHOS), we performed ChIP-qPCR to examine STAT3/pSTAT3 binding to the *Atpif1* promoter post AR signaling. We found that in untreated MDSCs, unphosphorylated STAT3 is attached to the *Atpif1* promoter. Following β -AR stimulation, STAT3 phosphorylation (pSTAT3) is enhanced leading to reduced STAT3 availability, decreasing binding to the *Atpif1* promoter and suppressing *Atpif1* expression. This is in accordance with the sustained stimulation of the STAT3 pathway at physiological levels in MDSCs relying on higher OXPHOS.”*

Comment 4: Itaconate is a well-established inhibitor of succinate dehydrogenase (SDH), which suppresses respiration. A concomitant increase in mitochondrial respiration and itaconate is difficult to understand and interpret in this context, again this is muddled by the experiment design. The combination of STAT3/ATP-itaconate doesn't represent an axis and this is unusual to suggest.

Response: *Thank you for the comment and recommendations. We conducted more experiments to show how itaconate affects mitochondrial respiration. To examine the TCA cycle metabolites and itaconate generation, we performed new tracing studies using $^{13}\text{C}_5$ -Glutamine. There are 4 groups of (WT + vehicle) vs (WT + ISO) and (β 2-AR KO + vehicle) vs (β 2-AR KO + ISO). Seahorse tracing results demonstrated enhancement of mitochondrial respiration, TCA cycle metabolites, and accumulation of citrate, cis-aconitate and itaconate post β -AR signaling (ISO treatment of WT MDSCs). The upregulated TCA cycle metabolites in MDSCs do not reflect the inhibition of SDH by itaconate as seen in LPS-activated macrophages (PMID: 27374498). This contrasts with recent findings of no effect of accumulated itaconate on SDH within tumor neutrophils (PMID: 37793345). A potential explanation for such discrepancies could be due to the difference in the cell-type-specific regulation of TCA cycle, difference in the cell activation strategies, and the kinetics of itaconate generation (discussed in PMID: 37793345). This recent publication (PMID: 37793345), establishes the role of itaconate generation in mitochondria for the control of neutrophil survival, similar to our MDSC findings. We added text to the Discussion (pages 13-14, lines 317-336 and pages 19-20, lines 481-495). We also removed “STAT-3/itaconate axis” from the manuscript.*

Comment 5: The tracing approach is also quite over complicated, not super informative especially because the lack of controls. It would have been informative to assess levels of these metabolites with unlabelled metabolomics (with the appropriate controls) and follow up with single-labelled

tracers (U-13C-glutamine for e.g.) to assess isotopologue distributions and differences in glutamine carbon incorporation into the TCA cycle.

Response: We agree and conducted the requested experiment using ¹³C-Glutamine tracing to increase the clarity of the results. We studied 4 groups: (WT + vehicle) vs (WT + ISO) and (β2-AR KO + vehicle) vs (β2-AR KO + ISO). The results are reported in diagrammatically in Figures 5A-B and Supplementary Figure 6 (S6) and with glycolysis changes in S7. TCA cycle metabolites for unlabeled (0), isotope labeled from glutamine (C*) and total (unlabeled + labeled) at 2 and 8 hr time points (effective isotope labeling timepoints) are presented. To prevent data overcrowding, the tracing of TCA cycle metabolites for (WT + vehicle) vs (WT + ISO) are presented in Figure 5. The (β2-AR KO + vehicle) vs (β2-AR KO + ISO) did not result in significant changes in TCA cycle metabolites (Figure S6). Metabolic changes in glycolysis are presented in Figure S7. We have added text to describe these results.(page 11, line 271-272; pages12-13, line 302-314; pages13-14, lines 317-326).

Comment 6: Acod1-/- in vitro data is weak/unconvincing so much of the itaconate angle relies on the itaconate ester 4-octyl itaconate (4-OI), which is a more potent NRF2 activator.

Response: Thank you for the comment and the recommendation. The results are statistically significant but subtle. Please also see our response to Reviewer 1, Comment 4. We have often discussed the lack of major effects from application of adrenergic stress. . We believe that the stress in these experiments would be reflective of chronic physiologic stress where the immune system is impaired at a mild level, but this occurs daily. While subtle, these effects can be cumulative, and likely be detrimental to long term immune surveillance. We hypothesize that if major effects were seen with chronic stress, then animals (or humans) would experience major, acute, life-threatening immune suppression from stressful events which can often occur.. Although the effects are subtle, they result in a statistically significant enhancement of immunosuppression, promoting tumor growth. Daily, chronic immunosuppression reduces the ability of doxorubicin to kill tumor cells more effectively. We have also better addressed this issue in the Discussion (page21, lines 513-518).

Reviewer #3 – sympathetic nervous system (B-adrenergic), cancer

No comments for Authors

Reviewer #4 - Blood cancer, metabolism, some sequencing (Remarks to the Author):

Major:

Comment 1: The conclusions are substantially weakened by the lack of proper controls. Because of this, most of the data are inconclusive as presented. Throughout the manuscript, various genetic mutants are used as “controls” for wildtype cells treated with drugs. Most notably WT + isoproterenol MDSCs are repeatedly compared to B2AR KO MDSCs without isoproterenol (Figs

2, 3, 4, 5). It is not clear why this comparison was chosen, and it seems likely that it will magnify most of the phenotypes. Importantly, the manuscript does not demonstrate that B2AR KO MDSCs develop or function normally. More appropriate comparisons would be (WT + vehicle) versus (WT + isoproterenol) versus (B2AR KO + vehicle) versus (B2AR KO + isoproterenol). Similar issue with Acod1 KO mice; for Fig 7K, what does survival look like in Acod1 KO + PBS?

Response: *We appreciate each of these comments and recommendations and fully agree with the reviewer (Please also see our responses to Reviewer 1, Comment 1 and Reviewer 2, Comment 2.) We initially reported the difference between WT and $\beta 2AR^{-/-}$ cells in vitro, to recapitulate and compare groups from the in vivo experiments (i.e., low vs. high β -AR signaling). We agree with the concern regarding controls. We have repeated the in vitro experiments adding more extensive control groups (WT + vehicle) vs (WT + ISO) vs ($\beta 2AR$ KO + vehicle) vs ($\beta 2AR$ KO + ISO). These are presented in Figures 2-5 and supplementary Figures S2, S4, S6-S9. In these experiments, we compared the β -AR activation (ISO) with WT to examine the effect of β -AR signaling and used ($\beta 2AR$ KO + vehicle) vs ($\beta 2AR$ KO + ISO) to control the specificity of the $\beta 2AR$ signaling. The new experiments support the role of the ATP/Itaconate pathway post- $\beta 2AR$ signaling in the control of MDSC mitochondrial fitness following doxorubicin treatment and tumor control in the EL4 lymphoma model.*

Other:

Comment 2: For all the in vivo experiments, it is not demonstrated (only inferred) that the effects on lymphoma survival are due to loss of MDSCs. It would help to show effect on survival with antibody or diphtheria toxin mediated depletion of myeloid cells, and further demonstrate that the various interventions have no effect on survival when MDSCs are not present. The exception to this is the experiments in Fig 1 with LysM Cre mediated deletion of B2AR which better narrow the effect to MDSCs.

Response: *Thank you for pointing this out. Please also see Reviewer 1 Comment 2 and our response. We repeated the in vivo experiments in Figure 1A (now 1A-B), Figure 7A (now 7A-B; line 412-419), and Figure 7L (now 7M; page 17-18, lines 434-438) using an anti-Gr1 MDSC depleting antibody to confirm the role of MDSCs in drug treatments. The administration of the anti-Gr1 MDSC depleting antibody blocked the suppressive function of 4OI treatment (Figure 7A-B; page 16-17, line 408-414). MDSC depletion by the anti-Gr1 antibody enhanced the metformin treatment effect, comparable to metformin+doxorubicin therapy (Figure 7M; pages 17-18, lines 434-438). These results suggest an important role for metformin to regulate the MDSC Electron transport chain (ETC) and tumor response. We also revised the text to discuss the possibility of direct drug effects and indirect effects of tumor burden on the MDSC percentage (page 19-20, lines 481-495 and pages 20-21, lines 502-518). We discussed that fact that $Lysm^{Cre}$ might also affect other myeloid cells including macrophages and conventional DCs (page 19, lines 462-468) as follows:*

“These data are consistent with a role for MDSC β -AR signaling and tumor growth control. However, there could be additional effects of propranolol on the tumor cells and other immune cells (57-59) modulating overall tumor growth outcome. Targeting MDSCs was accomplished by anti-Gr1 antibody treatment in the β 2-AR^{-/-} controls and by myeloid cell depletion using β 2-AR^{-/-} LysM^{Cre} mice. A limitation in this model is that targeting β 2-AR by LysM^{Cre} could potentially affect other myeloid cells including macrophages and conventional DCs”.

Comment 3: Metformin has many proposed mechanisms of action beyond complex I inhibition. For Fig 7, it should be demonstrated that metformin is inhibiting ETC function at the doses used. Likewise, effects of metformin on the relevant metabolic parameters should be shown (e.g. itaconate levels, ROS, etc) to better support the notion that it is acting through the proposed mechanism.

Response: *Thank you for this comment and we agree fully. We have now conducted experiments to investigate the role of metformin on ETC function and ROS levels (Figure S9; pages 17-18, lines 423-438). Our Seahorse and ATP production data suggest that the used doses of metformin indeed blocks ETC function, but the off target effects can not be ruled out. Therefore, to point out the potential off target effects of metformin beyond ETC inhibition, we have added the following to the discussion (pages 20-21, lines 502-518): Metformin treatment improved doxorubicin efficacy and with anti-Gr1 antibody treatment metformin alone was equivalent to metformin plus doxorubicin. Metformin plays a role in controlling MDSC ETC, thereby impeding the advancement of tumors. The impact of metformin on cancer management is currently under examination, with more than 400 studies registered on clinicaltrials.gov. It is crucial to delve deeper into understanding how metformin influences ETC regulation and other cellular functions. This includes exploring AMPK-dependent and AMPK-independent mechanisms, inhibiting mitochondrial respiration, inhibiting mitochondrial glycerophosphate dehydrogenase, and engaging mechanisms associated with the lysosome (68, 69).*

Some of the changes post β 2-AR stimulation are subtle resulting from chronic, daily, physiologic stress which impairs the immune system. These effects can be cumulative, resulting in a significant chronic, loss of tumor control. If major changes in immune function were seen daily, then experimental or clinical subjects experiencing chronic stress would experience major, acute, life-threatening immune suppression. Chronic immunosuppression reduces the ability of doxorubicin to effectively kill tumor cells to control tumor growth. Please also see our response to Reviewer 1, Comment 4 and Reviewer 2, Comment 6.

Comment 4: The single cell RNA sequencing data in Fig 2H and 3C is very difficult to decipher when presented in this way.

Response: *Thank you for the comment and request for clarification. We have repeated scRNA seq using new groups and the new data are presented as violin or bar graph plots (instead of heatmaps) which are easier to discern (Figure 2H: now 2G and in the Results: page 6, lines 146-149; Figure 3C; page 8, lines 180-181).*

Comment 5: For Fig 3E, it is not clear how this measure of JAK-STAT activation was performed. Should provide more details and/or show Western blot as well.

Response: We completely agree and thank you for the comment for clarification. We report the proteomic data obtained from the Phospho-Antibody Array kit (Full Moon BioSystems) results to generate Fig 3E (now Fig. 3D). We have provided the assay information and the protein signature list in the Figure legends and in the Materials and Methods (page30, lines 743-746). We also performed Western blot analysis to confirm STAT-3 phosphorylation in MDSCs after ISO treatment presented in Fig. 3E-F (page8, lines 192-198).

Comment 6: For Fig 3 experiments with JSI-124, it is not demonstrated that this is drug acting on target. Should be shown by western blot or better with genetic experiment targeting STAT3.

Response: Thank you for the observation and recommendation. We used western blot analysis to confirm on targets effect of JSI-124 on inhibition of STAT-3 phosphorylation shown in Figure 3E-F. These results are consistent with other studies demonstrating the on-target effect of JSI-124 (e.g. PMID: 12649187, PMID: 16939498, PMID: 23262510). We have clarified in Results section (page8, lines 192-198).

Comment 7: For Fig 3, the Atpif1 data is correlative and this should be more explicitly acknowledged in the text.

Response: Thank you for the observation and comment. (Please also see Reviewer 2, Comment 3 and the Response. We used the specific STAT3 inhibitor (JSI-124) to confirm function of STAT3 signaling on Atpif1 expression (Fig. 3K). Additionally, we performed ChIP-qPCR to confirm that STAT3/pSTAT3 binding to Atpif1 promotor post AR signaling regulates the Atpif1 expression (Fig. 3L). Unphosphorylated STAT3 (STAT3) binds to the Atpif1 promoter as a regulator of Atpif1 expression in untreated MDSCs. β 2-AR stimulation enhances STAT3 phosphorylation (pSTAT3) resulting in a reduced STAT3/pSTAT3 ratio (Fig. 3E-F). This leads to the lower availability of STAT3 to bind to the Atpif1 promoter causing a reduction in Atpif1 expression (Fig. 3L). Blockade of STAT3 phosphorylation by JSI-124 reverses this effect, leading to enhanced ATP1F1 expression (Fig. 3K). The data are presented in Figure 3E-L. The following explanation is added to the manuscript (pages 9-10, lines 221-242):

“STAT3 signaling is important for shaping MDSC differentiation and function. This includes regulating the gene expression profile toward a suppressive phenotype and functions. Although pSTAT3 regulates gene expression at early time points post activation, unphosphorylated STAT3 is responsible for pSTAT3 non-responding genes at later time points (PMID: 17510282). Unphosphorylated STAT3 accumulates in response to gp130-linked cytokines (e.g. IL-6), activating transcription. Unphosphorylated STAT3 sustains cytokine-dependent signaling at late times through a distinct mechanism from pSTAT3. To examine how the STAT3 pathway post β 2-AR signaling reduces Atpif1 expression (needed for MDSC-enhanced OXPHOS), we performed

ChIP-qPCR to examine STAT3/pSTAT3 binding to the Atpif1 promotor post AR signaling. We found that in untreated MDSCs, unphosphorylated STAT3 is attached to the Atpif1 promoter. Following β 2-AR stimulation, STAT3 phosphorylation (pSTAT3) is enhanced leading to reduced STAT3 availability, decreasing binding to the Atpif1 promoter and suppressing Atpif1 expression. This is in accordance with the sustained stimulation of the STAT3 pathway at physiological levels in MDSCs relying on higher OXPHOS.”

REVIEWERS' COMMENTS

Reviewer #1 (Remarks to the Author):

Authors made a concerted effort to address my comments and concerns. I believe their answers were satisfactory.

Reviewer #2 (Remarks to the Author):

Dear authors,

While it is commendable the efforts to conduct additional experiments there is still sufficient concern with the mechanistic details. There is no 'ATP/itaconate pathway' and a change of title is required. ATP1F1 data is still correlative and it should be stated that it may contribute but it is not proven. The importance of endogenous itaconate-NRF2 is also unconvincing and still heavily relies on 4-OI. It may be suggested that it could play a role but the modest *Acod1*^{-/-} data may not support this hypothesis. The manuscript message and framing needs to be reconsidered in this reviewers opinion.

Reviewer #4 (Remarks to the Author):

The authors have adequately addressed this reviewer's suggestions and concerns. The manuscript is improved and suitable for publication.

REVIEWERS' COMMENTS

Reviewer #1 (Remarks to the Author):

Comment: Authors made a concerted effort to address my comments and concerns. I believe their answers were satisfactory.

Response: We appreciate the reviewer's support.

Reviewer #2 (Remarks to the Author):

Dear authors,

Comment: While it is commendable the efforts to conduct additional experiments there is still sufficient concern with the mechanistic details. There is no 'ATP/itaconate pathway' and a change of title is required. ATP1F1 data is still correlative, and it should be stated that it may contribute but it is not proven. The importance of endogenous itaconate-NRF2 is also unconvincing and still heavily relies on 4-OI. It may be suggested that it could play a role but the modest Acod1-/- data may not support this hypothesis. The manuscript message and framing needs to be reconsidered in this reviewer's opinion.

Response: We appreciate the reviewer's positive comments. We have revised the manuscript according to the reviewer's comments that are listed below:

1- The title was revised to: "Myeloid-derived suppressor cell mitochondrial fitness governs chemotherapeutic efficacy in hematologic malignancies."

2- We revised the title, abstract, results and discussion to highlight the role of "mitochondrial fitness" in regulation of MDSCs function. We toned down claims regarding the underlying mechanism and used the STAT3-pathway, ATP and itaconate metabolism instead of 'ATP/itaconate pathway.'

We have made edits using the phrase STAT3-pathway, ATP and itaconate metabolism in Lines 90, 401, 403, 407, 448, 533, 540.

3- We revised the text regarding the ATP1F1 indicating the potential "Contribution" of ATP1F1 in the process. (Line: 248-251 and Line: 526-529).

4- We revised the manuscript text to discuss the limitations regarding the importance of endogenous itaconate-Nrf2 compared to 4OI treatments. (Line: 395-399 and Line: 529-531)

Reviewer #4 (Remarks to the Author):

Comment: The authors have adequately addressed this reviewer's suggestions and concerns. The manuscript is improved and suitable for publication.

Response: We appreciate the reviewer's support.